# MethylBERT enables read-level DNA methylation pattern identification and tumour deconvolution using a Transformer-based model

Yunhee Jeong [1] ✉, Clarissa Gerhäuser [1], Guido Sauter[2], Thorsten Schlomm [3], Karl Rohr[4] & Pavlo Lutsik [1,5] ✉

DNA methylation (DNAm) is a key epigenetic mark that shows profound alterations in cancer. Read-level methylomes enable more in-depth analyses, due to their broad genomic coverage and preservation of rare cell-type signals, compared to summarized data such as 450K/EPIC microarrays. Here, we propose MethylBERT, a Transformer-based model for read-level methylation pattern classification. MethylBERT identifies tumour-derived sequence reads based on their methylation patterns and local genomic sequence, and estimates tumour cell fractions within bulk samples. In our evaluation, Methyl-BERT outperforms existing deconvolution methods and demonstrates high accuracy regardless of methylation pattern complexity, read length and read coverage. Moreover, we show its applicability to cell-type deconvolution as well as non-invasive early cancer diagnostics using liquid biopsy samples. MethylBERT represents a significant advancement in read-level methylome analysis and enables accurate tumour purity estimation. The broad applicability of MethylBERT will enhance studies on both tumour and non-cancerous bulk methylomes.

DNA methylation (DNAm) refers to enzymatic modification of nucleotide bases in the DNA with methyl groups. In animal and many plant genomes, cytosine followed by guanine (CpG) is the most frequent methylation target. Aberrant methylation patterns at CpGs reflect epigenetic heterogeneity in human tumours[1]. Therefore, DNAm data has been widely used for estimating tumour purity to examine the tumour epigenetic landscape relevant to clinical outcomes, tumour diagnosis and phenotypic characteristics[2–4].

DNAm can be profiled by sequencing methods such as whole genome bisulfite sequencing (WGBS)[5] or reduced representation bisulfite sequencing[6], amplification-free long-read sequencing, e.g. using

Oxford Nanopore Technologies or Pacific Biosciences platforms, as well as with bisulfite-based microarray methods like Infinium 450K/EPIC arrays[7]. High-quality sequencing-based profiling produces sequence reads covering a broad range of genomic regions with sufficient read depth (30x being a de facto industry-wide standard), and thereby preserves single-molecule signals of rare cell populations[8]. Sequencing-based data is even more crucial in circulating tumour DNA (ctDNA) analysis that facilitates non-invasive early diagnosis, prognosis and treatment response monitoring in cancer patients[9–11]. Nevertheless, most purity estimation or cell-type deconvolution methods have been developed for array-based DNAm profiles because of the intuitive

[1]Division of Cancer Epigenomics, German Cancer Research Center (DKFZ), Heidelberg, Germany. [2]Institute for Pathology, University Medical Center Hamburg-Eppendorf, Hamburg, Germany. [3]Department of Urology, Charité – Universitätsmedizin Berlin, Berlin, Germany. [4]Biomedical Computer Vision Group, BioQuant, IPMB, Heidelberg University, Heidelberg, Germany. [5]Department of Oncology, KU Leuven, Leuven, Belgium. ✉e-mail: y.jeong@dkfz-heidelberg.de; pavlo.lutsik@kuleuven.be

application of classical linear algebraic algorithms to matrices of average DNAm levels (beta-values)[12]. Most ctDNA data analysis methods likewise use beta-values. Moreover, our previous analysis showed that existing sequencing-based deconvolution methods do not perform better than array-based methods implying that they fail to fully exploit the advantages of sequencing data for accurate inference[13].

To overcome these limitations, we propose MethylBERT, a deep learning method for read-level methylation pattern identification and tumour purity estimation based on Bidirectional Encoder Representations from Transformers (BERT)[14]. Methyl-BERT uses a modified BERT model to encode read-level methylomes and classifies tsequence reads into tumour or normal cell types. Resulting posterior probabilities of cell types are used to derive tumour purity estimates through Bayesian probability inversion and maximum likelihood estimation. Along with purity estimation, MethylBERT provides the model precision based on Fisher information and estimation adjustment taking region-wise tumour purity into account. The application of Transformers has been overlooked in sequencing-based tumour purity estimation, and MethylBERT suggests an approach to using Transformers for sequence read classification that is different from previous methods (Supplementary Table 1).

We have thoroughly evaluated MethylBERT and compared it with existing methods. The results demonstrate MethylBERT outperforms other methods in read-level methylation pattern classification and tumour purity estimation. In our evaluation, we not only investigate the performance of MethylBERT but also analyse what the model actually learns via pre-training using reference genome sequences. Moreover, we suggest using the Fisher information to measure the precision of the estimation model. This gives guidance about how accurate the estimated tumour purity is, which represents essential information for analysing bulk samples without ground-truth tumour purity.

Finally, we also propose using MethylBERT for early diagnosis of cancers based on ctDNA analysis as well as cell-type deconvolution.

## Results

### MethylBERT overview

MethylBERT includes three main steps (Fig. 1). First, the MethylBERT model is pre-trained with a reference genome processed into 3-mer sequences. After pre-training, the MethylBERT model is fine-tuned to learn the read-level methylation pattern classification task. The output converted with a softmax function is interpreted as the posterior probability $P(c_j|r_i)$ of the cell type $c_j$ given a read $r_i$. In our case, the cell type is either tumour ($T$) or normal ($N$). Classification is performed by assigning the cell type with a higher posterior probability.

For the final tumour purity estimation, we apply Bayes' theorem to compute the probability $P(r_i|c_j)$ in the likelihood function using $P(c_j|r_i)$ assuming that every read has an equal marginal probability. Afterwards, the tumour purity is determined by maximum likelihood estimation. The estimated tumour purity can also be adjusted based on the skewness of the region-wise tumour ratio. The adjustment is particularly useful when the analysed bulks have a very high or low ratio of tumour-derived reads. A detailed description of the three steps of MethylBERT is provided in Methods.

### MethylBERT classifies complex read-level methylation patterns

We simulated read-level methylomes in different scenarios to evaluate the robustness of the MethylBERT read classification with respect to the pattern complexity (the details are in Methods). We compared our method with CancerDetector[15] and DISMIR[16], as well as a baseline method implemented using a hidden Markov model (HMM) designed to classify read-level methylation patterns into cell types.

In our first scenario, we simulated different complexities of 150 bps read-level methylomes from a beta-binomial distribution. Our

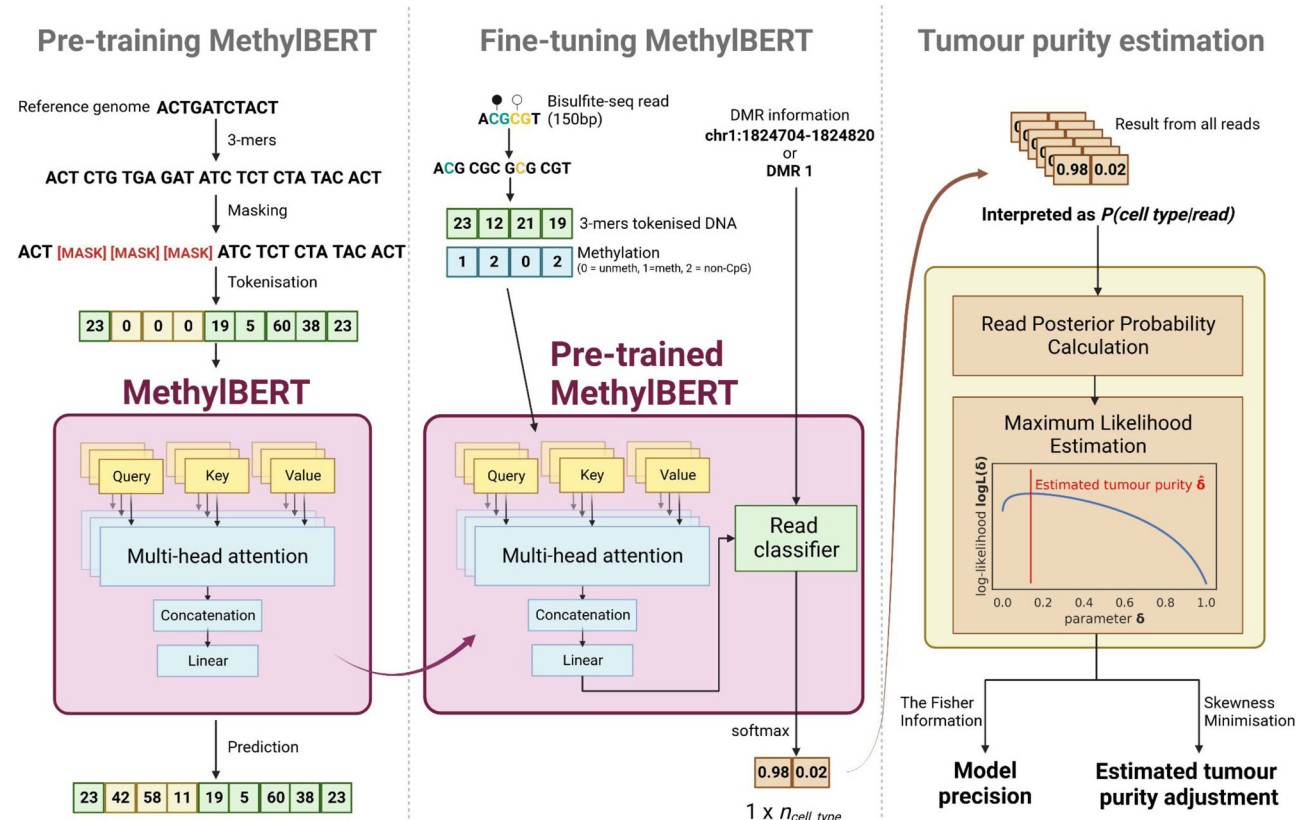

**Fig. 1 | MethylBERT overview.** The main three steps are separated by dotted lines.

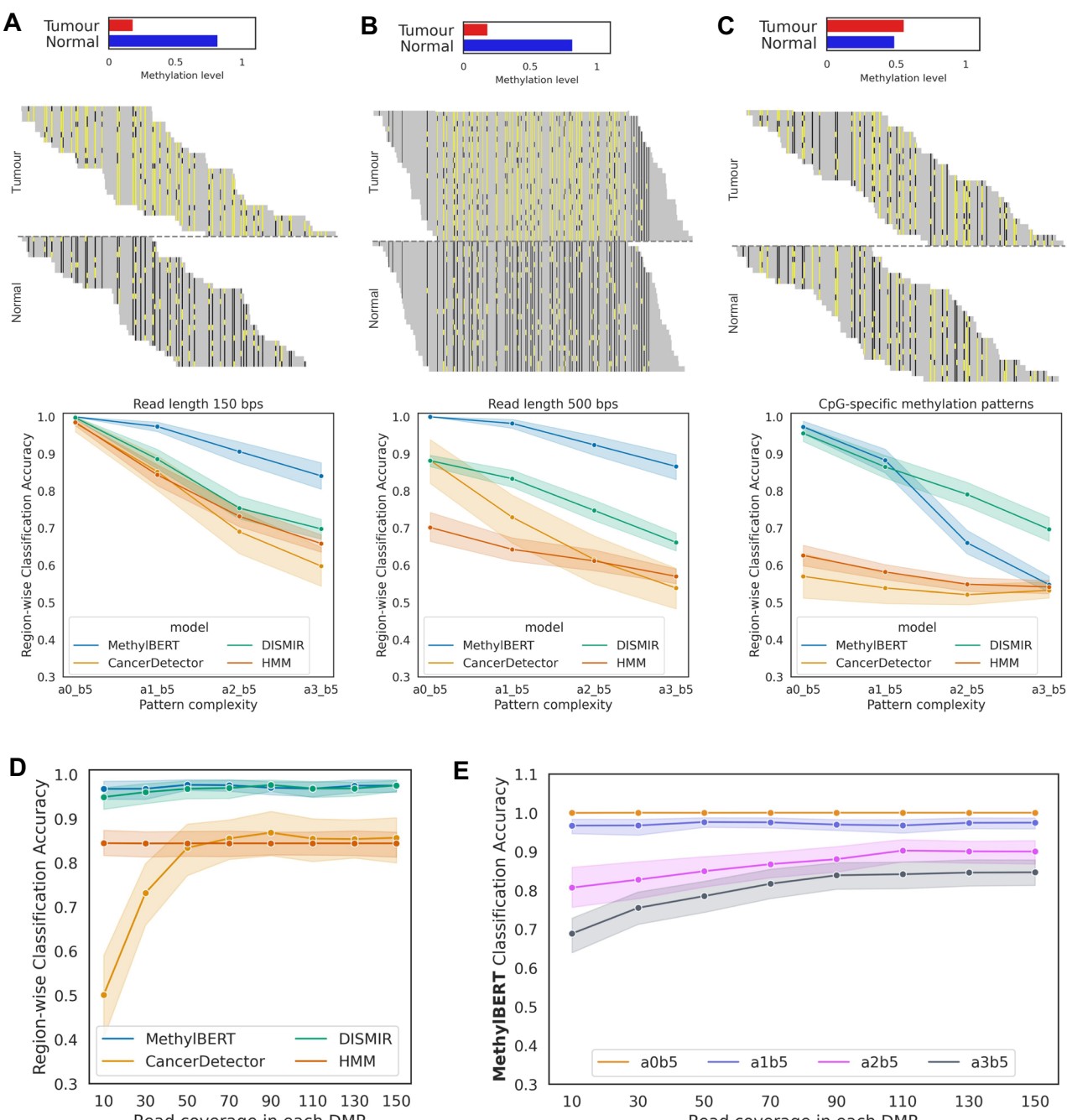

**Fig. 2 | Read-level methylome classification results.** In all line plots, the error bars indicate 95% confidence interval. **A**–**C** An example of simulated read-level methylation patterns for tumour and normal cell types (reads were simulated with the complexity of a1_b5, see Methods for details) and the classification accuracy comparison across different complexities (**A**) with read length 150 bps, (**B**) with read length 500 bps, and (**C**) with CpG-specific methylation patterns. Yellow and black on each read show methylated and unmethylated CpGs, whereas grey represents other bases. Sequence reads for two cell types are divided by a dotted line in the middle. Region-wise methylation levels in tumour and normal cell types are shown in the histogram. **D** Read-level methylome classification accuracy comparison for different read coverages. **E** MethylBERT read-level methylome classification accuracy for different read coverages and pattern complexities.

simulation algorithm generates more complex methylation patterns with an increasing $\alpha$ value of the beta distribution (Supplementary Fig. 2A, B). For all complexities of methylation patterns, MethylBERT outperformed the other three methods in the classification task (Fig. 2A). All four methods yielded the highest accuracy and the lowest deviation of accuracy values over differentially methylated regions (DMRs) for the simplest methylation patterns (a0_b5). Although the accuracy decreases with increasing complexity for all methods, MethylBERT still classified reads more accurately than the other methods in each case. We further simulated longer read-level

methylation patterns with 500 bp length. In this case, the reads cover both a DMR and non-DMRs on the genome resulting in more complicated methylation patterns, because most DMRs are shorter than 500 bps (Supplementary Fig. 2C). With the 500 bps of simulated reads, MethylBERT again performed better than the other methods regardless of the complexity (Fig. 2B). It likewise achieved more precise classification for simpler complexity keeping the lowest deviation of accuracy at the complexity a0_b5. On the other hand, for the same complexity level, CancerDetector and HMM performed worse and had a higher deviation of accuracy in the results with 500 bp reads than in

the results with 150 bp reads. Overall, the deep learning methods, MethylBERT and DISMIR, yielded better classification results for the 500 bp read simulation.

Neighbouring CpGs usually have a consistent methylation pattern, but the innate inclination of methylation for each CpG and erroneous methylation patterns in cancer are non-trivial factors[17,18]. Thus, we conducted the same evaluation for the extreme scenario where only CpG-specific methylation patterns differ between tumour and normal whereas region-specific methylation levels are almost identical (Methods, Supplementary Fig. 2D). CancerDetector and HMM could not classify reads into cell types despite the clear pattern difference in the case of a0_b5 and a1_b5, whereas MethylBERT performed best with the exception of highly noisy patterns in a2_b5 and a3_b5 (Fig. 2C). These results demonstrate that MethylBERT is capable of detecting tumour-specific methylation patterns not misled by the average methylation level in a region.

Read coverage is another important factor that can affect read-level methylome analysis. Many deconvolution algorithms pre-select regions based on the minimum number of CpGs and read coverage within the region so that sufficient methylomes can be used for deconvolution[13]. Hence, we simulated read-level methylomes with variable read coverages in DMRs. Again, MethylBERT achieved the best methylation pattern classification performance, especially for low read coverage (Fig. 2D). Although DISMIR showed competitive performance over different coverage values, we found that DISMIR training is less robust than MethylBERT training and partially yields low read classification accuracy (Supplementary Fig. 10). MethylBERT shows an accuracy value above 0.95 even for coverage 10 where the sample means cannot represent the population mean well (Supplementary Fig. 2E). On the other hand, CancerDetector could not perform accurate read classification for coverages below 50. We also performed the MethylBERT read classification analysis for every combination of read coverages and complexities (Fig. 2E). MethylBERT keeps a high accuracy regardless of the coverage for the complexity a0_b5 and a1_b5. However, the accuracy converges at the highest value for the read coverage > 100 in the complexity a2_b5 and a3_b5 results.

## Pre-training allows MethylBERT to understand sequence features

Bidirectional pre-training is a pivotal feature of the BERT model[14], alleviating the restricted choice of a model architecture. It was shown by Ji et al.[19] that the pre-trained BERT model can be successfully fine-tuned for various DNA sequence analyses such as promoter region prediction. Clark et al.[20] have carefully described what kind of information BERT learns during pre-training in natural language processing. Yet, the efficacy of BERT pre-training on DNA sequences is still poorly understood.

We have primarily found that pre-training enables the BERT model to understand the mutual relationships between DNA 3-mers (Fig. 3A). Even though any information about CpGs or methylation has not been provided, BERT distinguishes 3-mer tokens including "CG" from other tokens (cluster 3). This might be driven by the repetitive "CG" patterns especially occurring in CpG islands. In addition, the pre-trained BERT model is able to associate paired DNA nucleotides with each other (C-G and T-A). UMAP2 embedding divides the 3-mer tokens into two groups: those that start with C/T and the others starting with G/A. Each cluster is composed of tokens whose first nucleotide is the same and whose last nucleotide makes a nucleotide pair. For example, in cluster 6, all tokens start with A and end with C/T. We hypothesise that BERT pre-training identified the nucleotide pairs because of Chargaff's second parity rule that the amount of two nucleotides in a pair is approximately equal in a single DNA strand. Special tokens (e.g., <unk> or <mask>) also make up a separate cluster from the other DNA 3-mer tokens only after pre-training.

We have also evaluated the influence of pre-training on the identification of tumour methylation patterns. For this, we compared the performance of read classification when the MethylBERT model was pre-trained and when it was not pre-trained. The comparison was done by utilising diffuse large B cell lymphoma (DLBCL) and non-neoplastic B cell samples (Methods). In order to avoid the additional influence carried by the dominant promoter hypermethylation in the tumour, we selected 50 DMRs where tumour cells are hypermethylated and hypomethylated, respectively (Supplementary Fig. 3A, B). Both MethylBERT models with and without pre-training gradually decrease the loss value during the early steps of fine-tuning (Fig. 3C). However, when MethylBERT is not pre-trained, the loss value increases again after 100 steps and the accuracy eventually converges around 0.5. The confusion matrix of classified reads also shows that pre-trained MethylBERT achieves far more accurate classification results (Fig. 3D).

Figure 3E shows the change in the probability $P(cell\ type = Tumour|read)$ during MethylBERT fine-tuning. When MethylBERT is pre-trained with the reference genome, the probability distribution of tumour reads and normal reads start separating from each other at step 50. Over further steps, the model yields higher probability values of correct cell types for the reads. For instance, at step 350, $P(cell\ type = Tumour|read)$ of normal reads are close to 0 and the probability of tumour reads are close to 1. On the other hand, MethylBERT without pre-training could distinguish some tumour reads from normal reads until step 150, but afterwards the $P(cell\ type = Tumour|read)$ distribution of two cell types becomes indistinguishable. According to the correlation between the estimated probability and methylation level of reads, both models classify the reads mainly based on the methylation level at step 50 (Supplementary Fig. 3C). However, only the pre-trained model can overcome the bias and, in the following steps, keep the accuracy high without $P(cell\ type = Tumour|read)$ not being strongly correlated with the methylation level.

Pre-training on an entire reference genome requires a long training time, so the cross-species applicability of the pre-trained MethylBERT model will augment the utility to analyse various samples. For this reason, we have investigated the discrepancy in fine-tuning performance between pre-trained models with human (hg19) and mouse (mm10) genomes. The classification results of DLBCL and non-neoplastic B cell reads show that both human and mouse genomes are eligible as pre-training data for human cancer analysis (Fig. 3B). The read classification area under the curve score calculated in the validation set shows a difference below 0.001 between the two reference genomes. The distribution of calculated $P(cell\ type = Tumour|read)$ also does not significantly differ in non-neoplastic B cell reads. However, in the DLBCL reads, the two sets of probability values still have a $p$-value below $1.0 \times 10^{-5}$ for paired t-test statistics.

These results clearly demonstrate that pre-training is a vital step enabling the MethylBERT model to understand the major features of the DNA sequence. Furthermore, the MethylBERT fine-tuning can prevent bias towards read-wise methylation levels only when pre-training is performed. Finally, the cross-species applicability of the pre-training broadens the range of available samples for MethylBERT analysis.

## MethylBERT accurately estimates tumour purity of in silico bulk samples

We next evaluated the tumour purity estimation performance of MethylBERT using *in silico*-generated pseudo-bulk samples. The samples were obtained by mixing reads randomly sampled from DLBCL and non-neoplastic B cell samples with controlled proportions. For the comparison, we benchmarked MethylBERT against CancerDetector[15], DISMIR[16] and Houseman's method[21]. Houseman's method performed

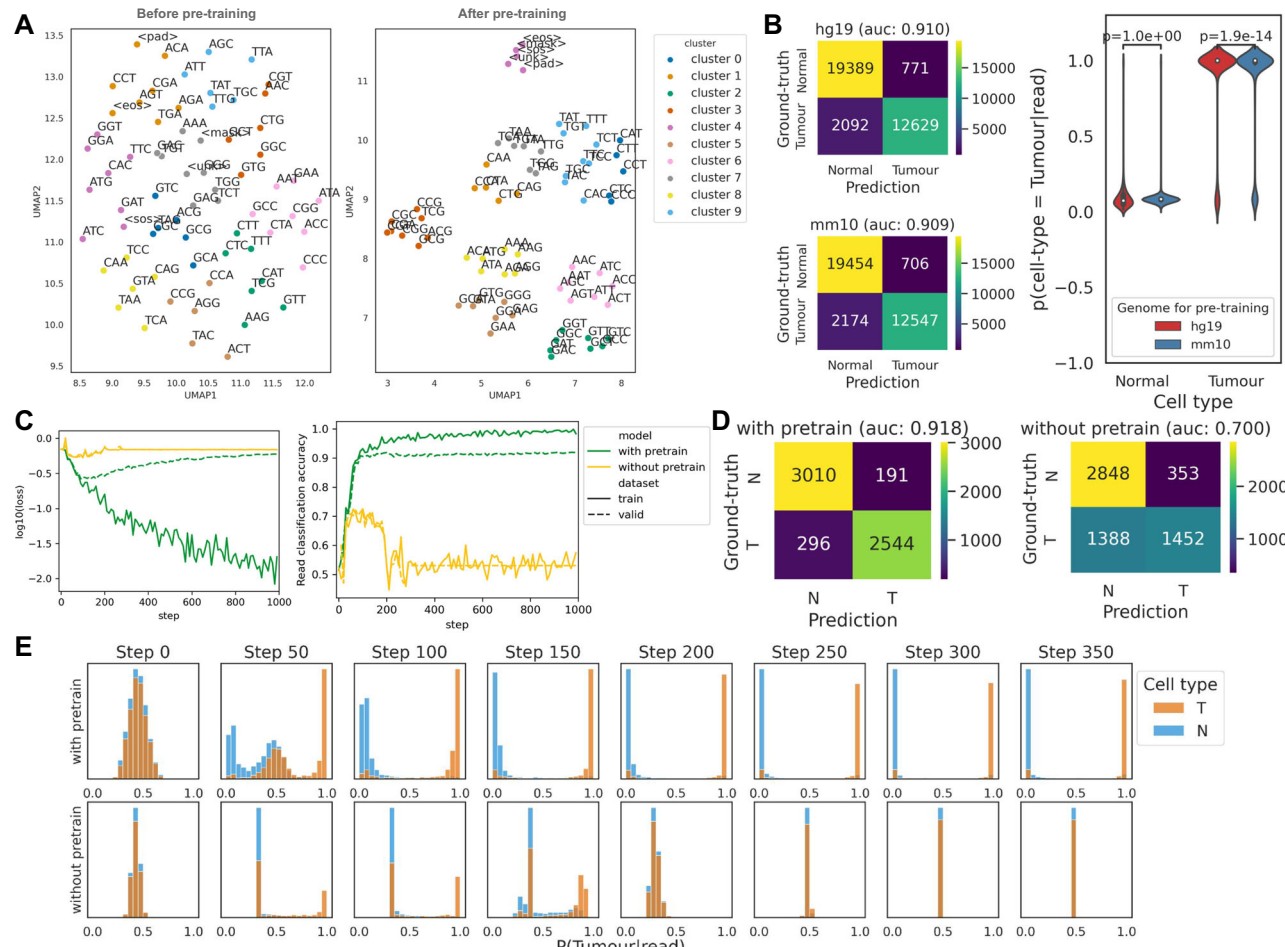

**Fig. 3 | Impact of pre-training on MethylBERT performance. A** UMAP plot of 3-mer token embeddings before and after pre-training. The clusters were made via *k*-means clustering. **B** Confusion matrix of read classification results by the model pre-trained with human genome hg19 (top left) and with mouse genome mm10 (bottom left). Distribution of *P(cell type = Tumour|read)* in both cell types calculated by the two pre-trained models (right). *P*-values in the violin plot were calculated using two-sided paired *t*-test statistics. The inner boxplots represent the median, and the first and third quartiles, whereas the whiskers show the rest of the distribution. **C** Training (solid line) and validation (dotted line) curves of Methyl-BERT with and without pre-training (green and yellow). Both graphs are plotted every 10 steps. **D** Confusion matrix of read classification results by the MethylBERT model with and without pre-training calculated at the step when each model achieved the best validation performance. **E** Histogram of *P(cell Type=Tumour|read)* in tumour (T) and normal (N) reads (orange and blue each) calculated by Methyl-BERT with and without pre-training (top and bottom).

best in diverse deconvolution and tumour purity estimation experiments in our previous benchmarking study[13]. Although DISMIR features a procedure for informative genomic region selection for tumour purity estimation, it could not make reasonable estimates with self-selected regions. Therefore, we applied DISMIR on the same DMRs used for the other methods and this is referred to as 'DISMIR_dmr' in the following. For the MethylBERT results, we present the performance from both models with and without estimation adjustment (described in Methods) to study the impact of the adjustment.

MethylBERT outperformed other methods with respect to the absolute error between the ground-truth and estimated tumour purity (Fig. 4A, Supplementary Table 2 and Supplementary Fig. 11). Although Houseman's method achieved better performance for the bulks with a high tumour purity, it could not accurately estimate the purity when the ground-truth value is low. On the contrary, CancerDetector performed better for the bulk samples where the tumour was a minor cell type. However, MethylBERT maintained its high accuracy for both low and high tumour purities.

To further improve the accuracy of tumour purity estimation, MethylBERT employs estimation adjustment based on the distribution of region-wise (local) estimated tumour purity (Methods). This is

conceptually equal to the 'removal of confounding factors' in CancerDetector[15], as it also handles outlier regions where the estimated proportion of tumour-derived reads is different from the majority. MethylBERT finds the optimal mapping of local estimates to reduce the skewness, whereas CancerDetector iteratively removes regions that are outside of the standard deviation. Therefore, we have specifically compared the MethylBERT estimation adjustment and CancerDetector removal of confounding factors using the same estimated *P(cell type|read)* values (Fig. 4B). The 'no adjustment' label indicates when neither of the adjustment methods was applied and the final tumour purity estimation was calculated only based on the *P(cell type|read)* values. MethylBERT estimation adjustment outperformed CancerDetector removal of confounding factors in terms of median absolute error between the ground-truth and estimated purities. Although CancerDetector removal of confounding factors cannot improve the estimated values when the tumour is the major cell type, MethylBERT is able to make a better adjustment regardless of ground-truth tumour purity (Supplementary Fig. 12).

In real application scenarios, ground-truth tumour purity is not available for evaluation. Therefore, as a quality measure of model estimates, MethylBERT provides the precision of the tumour purity

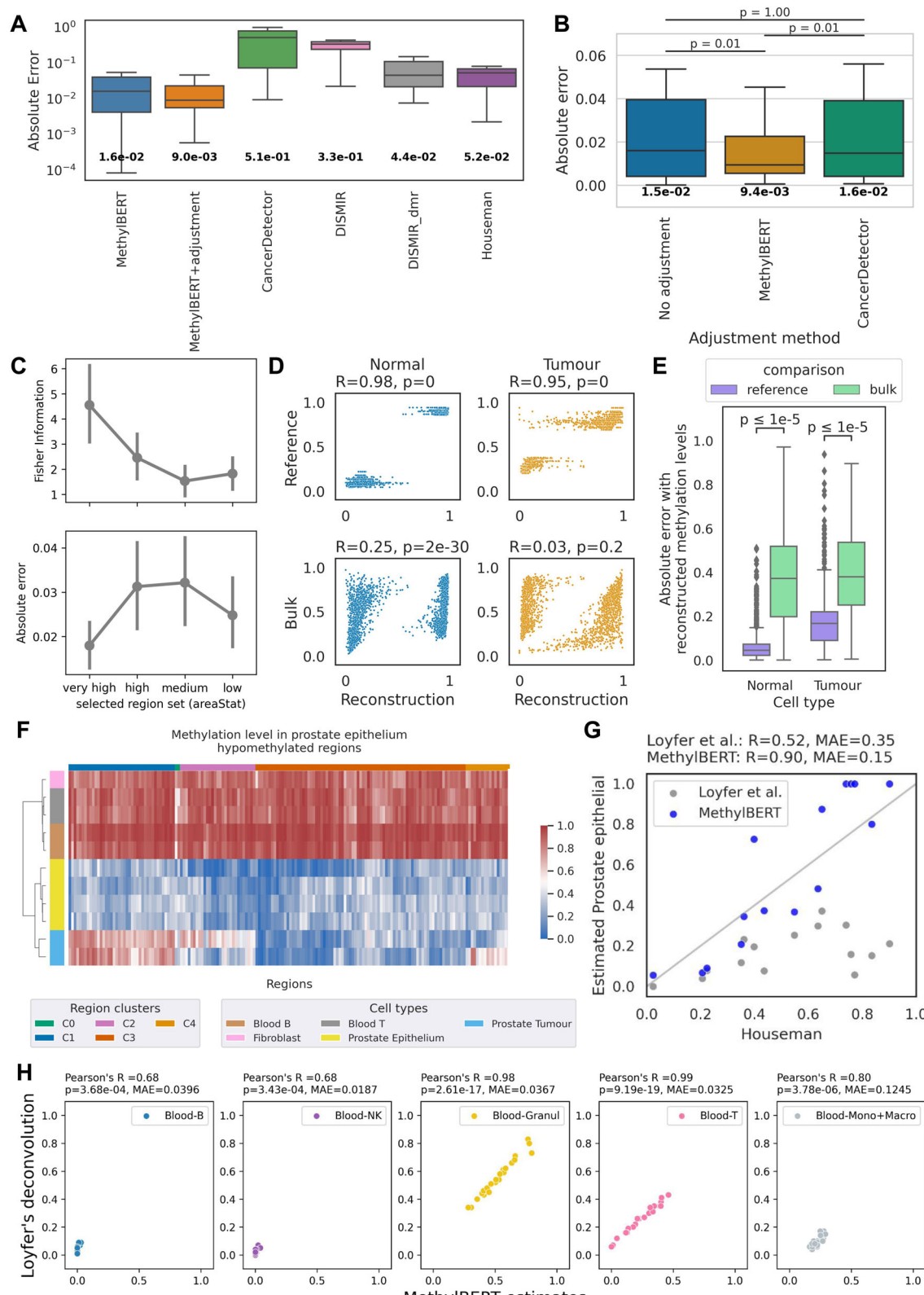

estimation using the Fisher information. When multiple experiments are performed by MethylBERT, users can obtain information on which experiment is more likely to make a precise estimation by comparing the Fisher information indicating the precision of likelihood estimation models[22]. Here, we show how the Fisher information can be used for determining the best DMRs. For this, we split the 100 DMRs ordered by the region quality (indicated by areaStat value as explained in

Methods) into four groups: very high, high, medium, and low, meaning that a higher areaStat value represents higher methylation difference and more CpGs within the region. Both tumour purity and the Fisher information were calculated in respective groups by MethylBERT (Fig. 4C). The result shows that the mean absolute error and the estimation quality measured by the Fisher information are anti-correlated. Hence, the Fisher information from MethylBERT can be used to analyse

**Fig. 4 | MethylBERT analysis results for bulk samples.** All boxplots represent the median, the first and third quartiles, whereas the whiskers show the rest of the distribution. **A**–**C** Tumour purity estimation and estimation adjustment results for DLBCL pseudo-bulk samples ($n = 20$). **A** Absolute error between the ground-truth and estimated tumour purity. **B** Performance comparison of different estimation adjustment methods. The boxplot presents the absolute error. The number at the bottom indicates the median value in both (**A**) and (**B**). **C** Fisher Information values and mean absolute error calculated for the pseudo-bulk samples with four different region sets. Dots and lines indicate the mean value and the confidence interval (95%) in each region set. **D**–**E** Cell type-specific methylation level reconstruction within 100 DMRs for 20 DLBCL pseudo-bulks ($n = 2000$). **D** Correlation calculated with reference cell type-specific methylation level (top), and calculated with bulk methylation level (bottom). **E** Absolute error calculated with reference methylation levels (purple), and calculated with bulk methylation levels (green). In (**B**) and (**E**)

statistics were computed using a two-sided paired $t$-test with Bonferroni correction. **F** Methylation level comparison between prostate tumour, normal prostate epithelium, T cell, B cell, and fibroblasts in the 250 regions provided by the normal cell-type methylation atlas[23]. Regions were clustered using the hierarchical clustering algorithm. **G** Correlation between the estimated prostate epithelium proportion (without tumour reference data) and estimated tumour purity (with tumour reference data) by Houseman's method for the lymph node samples ($n = 15$) acquired from hormone-sensitive metastatic prostate cancer patients. The results for MethylBERT and Loyfer et al.'s deconvolution methods are coloured blue and grey, respectively. **H** Cell-type deconvolution results for leukocyte samples between MethylBERT and Loyfer et al.'s method. In (**D**) and (**H**) the two-sided $p$-value represents the probability that the absolute correlation coefficient of a random sample from an uncorrelated population is greater than the absolute value of a given correlation coefficient, as implemented in the Scipy package[48].

the quality of maximum likelihood estimation when other information for the quality evaluation is not available.

In addition to the accurate inference of mixture proportions by MethylBERT, the read classification results can be used to reconstruct region-wise methylation levels of constituent cell types within bulk samples. From the classified reads, we calculated the average methylation levels of tumour and normal per DMR (so-called reconstructed methylation level) in each pseudo-bulk sample. Figure 4D, E present that the reconstructed methylation level is much more similar to the reference cell type-specific methylation level than to the bulk methylation level with a higher Pearson correlation and a lower mean absolute error. Bulk-wise reconstructed pattern analyses are shown in Supplementary Figs. 5A and 13. The results confirm that reconstructed values have a lower error with respect to the reference methylation patterns than with respect to the bulk patterns. Please note that, for testing, we used 100 regions where half are tumour hypermethylated and the other half are hypomethylated regions to make sure that the result is not influenced by dominant tumour hypermethylation over the selected DMRs based on the areaStat score. We performed the same analyses for the DMRs selected based on the areaStat score (Supplementary Figs. 5B, C and 14). MethylBERT still successfully dissected cell type-specific methylation levels showing the same results.

### MethylBERT facilitates cancer patient analyses and cell-type deconvolution using the methylation atlas of normal cells

When MethylBERT is used for practical applications, it could be the case that tumour reference sequencing data is not available. Thus, using lymph node samples collected from hormone-sensitive metastatic prostate cancer patients, we evaluated MethylBERT's applicability to analysing tumour bulk samples without cancer-derived reference data. Instead of tumour reference data, we used the normal cell-type methylation atlas data which includes blood cells and prostate epithelium cells[23]. We hypothesised that the estimation of prostate epithelium cell fraction should align with the tumour purity in lymph node samples because of tissue invasion and metastasis, which is a well-known cancer hallmark[24]. Therefore, we evaluated the MethylBERT-estimated prostate epithelial cell proportions compared to the prostate tumour purity estimated using prostate cancer reference data (as described in Methods). Despite the methylation level difference in some regions between normal prostate epithelium and prostate tumour samples (region clusters C1, C2 and C4 in Fig. 4F), the prostate-derived read proportion estimation by MethylBERT without tumour reference data shows a strong positive correlation with the prostate tumour purity estimated by other methods[21,25–27] trained with tumour reference data (Fig. 4G and Supplementary Fig. 4). We deconvolved the same samples with the UXM fragment-level deconvolution algorithm from the atlas study[23] (denoted as Loyfer et al.'s method), however, the results do not show as strong correlation. This application presents a practical use case of MethylBERT combined

with the atlas data[23] even when the reference data is only partially available.

The design of the MethylBERT model and the cell-type proportion estimation likelihood function are technically not limited to tumour methylomes. Therefore, we extended MethylBERT to cell-type proportion estimates beyond tumour and applied it to a cell-type deconvolution for 23 leukocyte bulk samples acquired from the atlas data[23] (Methods). Figure 4H shows the estimated cell-type proportions in leukocyte bulk samples compared to the estimates by the UXM fragment-level deconvolution algorithm from the atlas study[23]. For the five major cell types (B, NK, granulocytes, T and monocytes + macrophages) in the bulk samples, MethylBERT estimates have a strong correlation with the fragment-level deconvolution results. This confirms the applicability of MethylBERT to more complex cell-type deconvolution analyses for non-tumour bulk samples.

### MethylBERT accurately detects rare tumour signals in liquid biopsy samples from cancer patients

Besides bulk tumour analyses, precise estimation of sequencing-based tumour cell fraction is required in ctDNA analysis due to the very low quantity of tumour-derived DNA found in liquid biopsies at early disease stages (<5%). Hence, we evaluated MethylBERT as an early cancer detection method for blood plasma samples and compared its performance to other previous methods.

To validate whether MethylBERT can detect a very low percentage of tumour-specific signals, we further generated 10 pseudo-bulks by mixing reads from non-neoplastic B-cell and DLBCL samples with a tumour ratio smaller than 10%. MethylBERT achieved a lower median absolute error than the other methods (Fig. 5A and Supplementary Table 3). All methods can estimate tumour purities proportional to the ground-truth values with the exception of DISMIR applied with its own selected regions but MethylBERT shows the highest correlation between the two values with $p$-value < 0.01 (Fig. 5B). Therefore, we demonstrated that MethylBERT is sufficiently sensitive for ctDNA analysis compared to the currently available methods.

As an application to real ctDNA samples, we used targeted BS-seq data collected from 14 healthy donors and 40 colorectal cancer (CRC) patients in five different stages (GSE149438) (Supplementary Fig. 6A). The estimated tumour purity significantly differs between healthy donors and tumour patients later than stage I (Fig. 5C). This implies that MethylBERT can be instrumental for ctDNA tumour diagnosis in some early stages (II-III) of CRC patients. The median value of estimated tumour contents has an explicit discrepancy between the healthy donors and all stages of CRC patients.

From the same dataset, we also collected 44 pancreatic ductal adenocarcinoma (PDAC) patients in four different stages (IIA, IIB, III and IV) and conducted the same analysis using MethylBERT (Supplementary Figs. 6B and 5D). PDAC is widely recognised as one of the trickiest cancer types to be identified during the early stages, which is also confirmed by the original analysis in this dataset[28]. The median

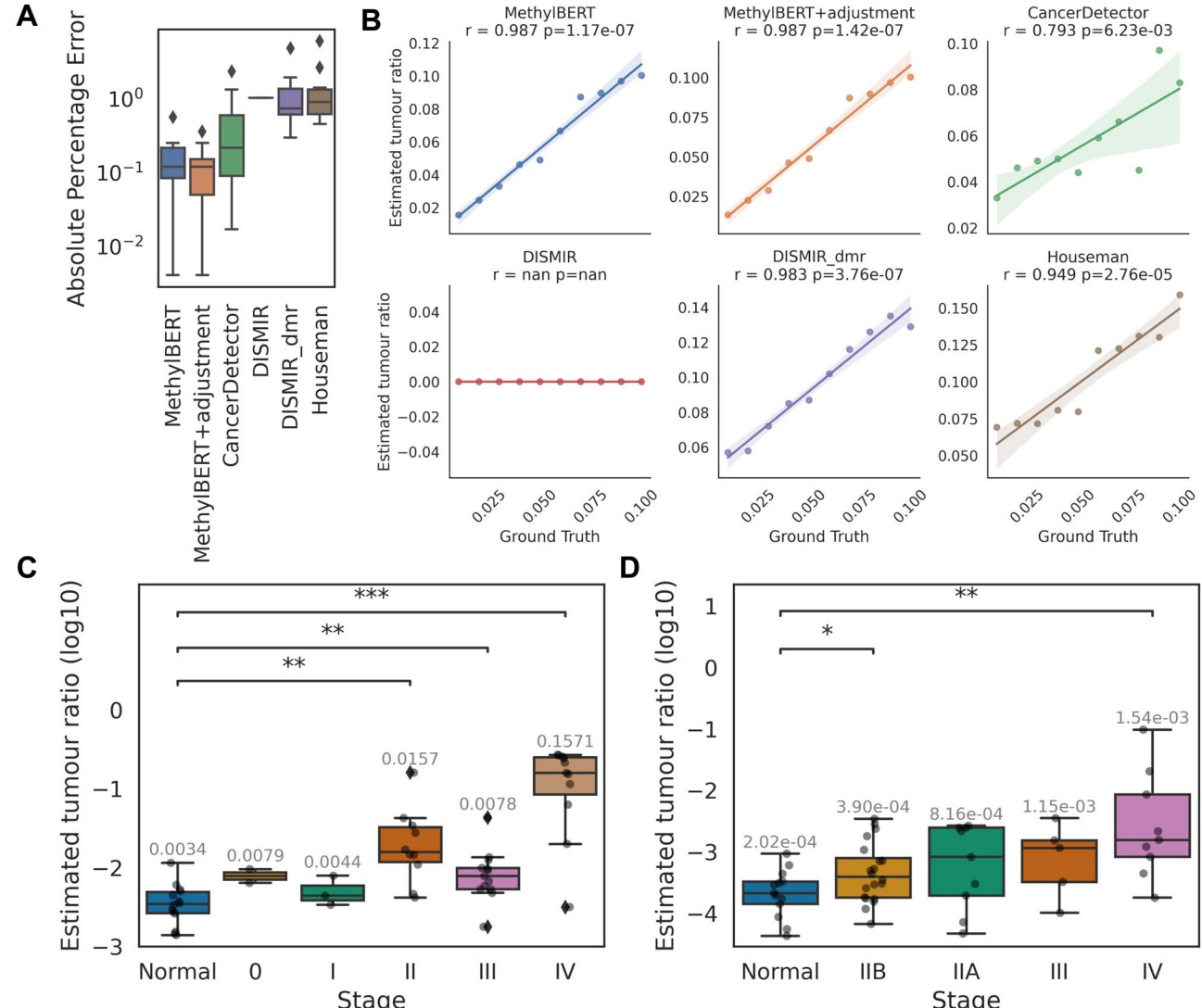

**Fig. 5 | Estimation of tumour fraction in plasma cell-free DNA.** All boxplots represent the median, the first and third quartiles, whereas the whiskers show the rest of the distribution. **A** Tumour purity estimation of simulated pseudo-bulk samples (*n* = 10) with a low percentage of tumour DNA. Distribution of absolute percentage error values in each method. **B** Comparison of ground-truth and estimated tumour purities in each method. For each comparison, Spearman's correlation is given with a respective *p*-value. The error bars indicate a 95% confidence interval. The two-sided *p*-values were calculated with a null hypothesis that two

samples do not have an ordinal correlation, as implemented in the Scipy package[48]. **C**, **D** Tumour cell fraction estimation results in ctDNA samples (**C**) from CRC patients and (**D**) PDAC patients (Supplementary Fig. 6 provides the sample size information). Both analyses include healthy donors as well. The median of estimated tumour purities in each stage is written at the top of the box plot. Two-sided Mann–Whitney–Wilxcoxon test *p*-values between each stage and healthy donors are denoted with stars. '*', '**' and '***' mean *p*-value ≤ 0.05, ≤ 0.01, and ≤ 0.001, respectively.

tumour purity value estimated by MethylBERT is higher in every stage of PDAC cohorts than in healthy donors. Among the early-stage patients, stage IIB showed a statistically significant difference from the healthy donors. Overall, in both CRC and PDAC ctDNA analyses, all early-stage cancer patient samples except for CRC stage II have a median value lower than 0.01 demonstrating the necessity of sensitive sequencing-based tumour cell fraction estimation models in ctDNA methylation analysis.

## Discussion

DNA methylation undergoes profound changes during tumorigenesis[1,29] resulting in highly specific methylation patterns in tumour cells. Sequencing data is particularly valuable in this respect, since it offers DNAm patterns at single-molecule resolution, broad genomic coverage and the preservation of rare cell-type signals.

Aiming to fully utilise the potential of sequencing-based data, we developed MethylBERT, a Transformer-based model for read-level

tumour methylation pattern identification. Based on the estimated posterior probability by the MethylBERT model, the tumour purity of each bulk sample can be inferred by maximum likelihood estimation. The estimated purity can be adjusted for a more accurate inference by considering region-wise tumour purity estimation.

In the benchmarking using simulated read-level methylation patterns with different scenarios, MethylBERT achieved the best performance regardless of methylation pattern complexity, read length and read coverage. Our benchmarking results demonstrate not only the accurate read classification and tumour purity estimation performed by MethylBERT in varying experiments but also provide in-depth analyses of BERT pre-training and the dynamics of the estimated posterior probability during model training. Moreover, MethylBERT is capable of deconvolving bulk samples into multiple cell types as well as distinguishing some early-stage cancer patients from healthy donors using blood plasma samples. This assures the applicability of MethylBERT as a cell-type deconvolution model as well as in the context of

early cancer detection. We are convinced that MethylBERT will become a valuable tool in the field of cancer research and oncology, extending the usability of various types of DNAm sequencing data. MethylBERT is agnostic with respect to the source of read-level methylation data, and can be applied to both solid tumour tissues and blood plasma samples as well as non-cancerous bulk methylomes, unlike previous methods that focused on only specific sample types[15,16,30].

In the future, different extensions of MethylBERT are possible. We are currently working on a computationally more efficient version of the MethylBERT model with a reduced number of parameters. This model optimization is particularly important for applications with long-read sequencing data[31,32]. Based on the results of read-level methylation classification with 500bp-long simulated reads, we believe that MethylBERT can handle differentially methylated patterns being a part of a read-level sequence, and will facilitate accurate and robust analyses of tumour-specific methylation patterns in long reads.

# Methods

## MethylBERT model

The aim of MethylBERT is to classify sequence reads with their CpG methylation patterns and sequence into dichotomous cell-type-related classes, e.g., tumour or normal. Applied to a set of genomic regions, it also provides a global maximum likelihood estimate of cell-type proportions for the two classes, e.g., tumour purity in the tumour-normal deconvolution scenario.

**BERT model.** The basis of MethylBERT is the BERT model which has achieved groundbreaking progress in various natural language processing studies[33,34]. Transformers[35] in the BERT model take the attention mechanism resembling human cognitive attention and map given query ($Q$), key ($K$) and value ($V$) sequences to an output sequence. Transformers particularly use scaled dot-product attention, where the dot-product of query and key sequences are scaled by the inverse square root of the dimension of key vectors ($d_k$):

$$Attention(Q, K, V) = softmax\left(\frac{QK^T}{\sqrt{d_k}}\right)V. \qquad (1)$$

The scale factor enables the model to avoid an extremely large magnitude of multiplied value which especially occurs with a long sequence input. Multi-head attention which concatenates $H$ multiple attentions calculated from the weighted query, key and value sequences is a major advantage of Transformers in learning different projections of these three sequences:

$$A_i = Attention_i\left(QW^Q_i, KW^K_i, VW^V_i\right), \qquad (2)$$

$$Multi - head\ attention(Q, K, V) = Concatenation(A_1, ..., A_H)W^O \qquad (3)$$

where $A_i$ refers to an attention head. In every attention head, given query, key and value matrices are projected using the parameter matrices $W^Q_i$, $W^K_i$ and $W^V_i$. Then, multi-head attention is created as a concatenation of all $H$ attention heads projected using another parameter matrix $W^O$.

We modified the BERT model to process DNA sequence fragments and sequential methylation patterns (Supplementary Fig. 1). Originally, BERT needs three types of input embeddings: token embeddings, segment embeddings and position embeddings. In MethylBERT, we replaced the token embeddings assigned for tokenised words with DNA token embeddings. The segment embeddings indicate the sentence label of each token and are used for next sentence prediction (NSP). Instead, for MethylBERT, we created methylation embeddings for encoded methylation patterns. The position embeddings were also used in MethylBERT to guide the position of tokens in the sequence read.

**Pre-training methylBERT.** As described above, BERT requires long pre-training to learn the general context of input data and avoids heavily engineered task-specific architectures for every specific task. We use the Masked Language Model (MLM) for pre-training while the original BERT method performs pre-training of the model via both MLM and NSP[14]. NSP is known to make the BERT model understand semantic dependencies across sentences[36], however, since we focus on the methylation pattern of unpaired reads, NSP was disregarded for MethylBERT pre-training. The overall pre-training scheme to use 3-mers for MLM is inspired by DNABERT[19].

We split the hg19 genome into 510 bp segments and generated a 3-mer sequence of each segment, and the 3-mer segment is referred to as a token. While Ji et al. randomly sampled the read length between 5 and 510 with a certain probability for DNABERT[19], we used a fixed value of the sequence length 510 since it did not make a major performance change. For pre-training, we only applied a MLM by randomly masking 15% of 3-mers. Three masking schemes were employed following the original BERT paper[14]: 80% of selected 3-mers were masked with [MASK] token, 10% were replaced with another randomly chosen token, and the rest was unchanged. We masked left and right tokens together with the selected token in order to prevent a biased model predicting the masked token from neighbouring tokens. 3-mers tokens have 69 labels in total including five special tokens. A categorical cross-entropy loss $L_{pre-training}$ was calculated over all tokens $t \in \{1...T\}$ for every step of the model optimisation:

$$L_{pre-training} = -\sum_{t=1}^{T}\sum_{l=1}^{69} m^t y^t_l \log(\hat{y}^t_l) \qquad (4)$$

where $y^t_l$ and $\hat{y}^t_l$ refer to as one-hot encoded value of label $l$ in token $t$ and corresponding logit value calculated by the MethylBERT model. $m^t$ is a binary value indicating whether token $t$ is masked or not.

We pre-trained the MethylBERT model for 120 k steps including 10 k warm-up steps and 20 k decrease steps at the end of the training with a learning rate of $4e^{-4}$. The batch size was set to 256 and the gradient was accumulated over 4 steps. In the network architecture, the hidden layer had a size of 768 and the baseline model had 12 encoder layers with 12 attention heads. However, since the smaller size BERT models performed similarly to the baseline model in the read classification analysis using simulated data, we used 6 encoder layers of the MethylBERT model for 500 bps read analysis (Supplementary Fig. 7). AdamW optimiser[37] was used with decay rate 0.01, $\beta_1 = 0.9$ and $\beta_2 = 0.98$ values. Since reference genomes do not have methylation patterns, we filled the methylation embeddings up with zeros during pre-training.

**Fine-tuning for read-level methylation pattern classification.** For fine-tuning, the encoder network in MethylBERT encodes a read-level reference DNA sequence and CpG methylation patterns (Supplementary Fig. 1). Although reference DNA sequences do not contain any tumour-specific genetic information, they carry the exact position of a read in a given region as well as sequence features associated with specific methylation patterns. This information is necessary for the model to learn which CpGs are more likely to show tumour-specific signals. Previous studies also have pointed out that integrated information of DNA sequences and methylation can improve modelling[16,38].

DNA sequence fragments from reads are processed into 3-mer sequences as described in the previous section. The input length was reduced to 150 due to the shorter sequence read length. In order to represent CpG methylomes, three numbers were used for methylation pattern encoding: 0 for unmethylated CpGs, 1 for methylated CpGs,

and 2 for non-CpGs. CpG methylation patterns were assigned to the 3-mers where the cytosine of CpG is located in the middle.

The encoder part of the MethylBERT network takes DNA, methylation and position embeddings, and generates an encoded vector of $sequence\_length \times 768$ dimension. This vector is concatenated with DMR information embedded into a space of dimension $sequence\_length$, and the following the cell-type classifier calculates the posterior probabilities of cell types using the concatenated vectors. The DMR label is provided as information to determine region-wise tumour-specific methylome profile. The cell type with the highest posterior probability is determined as the classification outcome.

The MethylBERT model was fine-tuned over 600 or 1000 steps with a learning rate of $4e^{-4}$. We used the same optimisation scheme as for pre-training but applied the cross-entropy loss on the cell-type label for read-level methylome classification:

$$L_{fine-tuning} = -\sum_{r=1}^{R}\sum_{c\in\{T,N\}} m_{r,c} \log \frac{\exp(x^c_r)}{\sum_{c'\in\{T,N\}}\exp(x^{c'}_r)} \quad (5)$$

where $x^c_r$ is the final activation for the cell type $c$ ($T$=tumour, $N$=normal) for each read $r$. $m_{r,c}$ is a binary value from one-hot encoded cell-type label c. The logits are normalised with a softmax function before the cross-entropy loss is calculated.

To understand the time complexity of MethylBERT fine-tuning, we provide the running time of MethylBERT by the number of encoder layers and number of GPUs in Supplementary Fig. 9. Using 541,000 reads for the training set and 135,000 reads for the validation set with a batch size of 600, four GPUs (Nvidia V100 SXM2 32GB) achieved the best time performance. However, we note that deep neural network training highly depends on the batch size set up and hardware specifications. For instance, increasing batch size could reduce the running time for 6–8 GPUs in Supplementary Fig. 9. This will result in faster training for the case of using 6–8 GPUs than using 4 GPUs, thus a larger batch size is recommended when users have a sufficient number of GPUs. Moreover, after one-time fine-tuning, the trained MethylBERT can be used for tumour purity estimation of several bulks independently and this takes a much shorter time. For instance, the tumour cell fraction estimation for one CRC ctDNA sample needed only less than 5 min with one GPU.

## Tumour purity/fraction estimation

Since the proportions of the tumour and non-tumour compartments sum up to one, we used a single-parameter likelihood function to estimate the best tumour purity $\hat{\delta}$ from collected reads $\{r_1, ..., r_N\}$:

$$L(\delta) = \prod_{i=1}^{N}\left[\delta P(r_i|cell\,type = Tumour) + (1-\delta)P(r_i|cell\,type = Normal)\right], \quad (6)$$

$$\hat{\delta} = argmax_\delta L(\delta). \quad (7)$$

The MethylBERT model calculates only the posterior probabilities of cell types given a read, so Bayes' theorem is applied to calculate the posterior probability of a read assuming that every read has the same marginal probability:

$$P(r_i|cell\,type = T) \propto P(cell\,type = T|r_i)P(cell\,type = T)^{-1} \quad (8)$$

The prior probability of the cell types is calculated from the training dataset used for fine-tuning. For less complex likelihood computation and the Fisher information calculation, we use the log-likelihood function for maximum likelihood estimation. We employ a grid-search algorithm to find the optimal parameter $\hat{\delta}$ increasing the $\delta$ value by 0.0001 from zero to one.

As shown in Supplementary Fig. 8A, the ratio of tumour-derived reads in DMRs does not have a symmetric distribution when tumour-normal cell types do not have an equal proportion. This is also shown by the negative correlation between the ground-truth tumour purity and the skewness of region-wise tumour purities (Supplementary Fig. 8B). However, estimating the tumour purity only using the log-likelihood function above assumes that tumour-derived reads are equally distributed in every DMR. Therefore, we propose an adjustment of estimated tumour purity to take the region-wise tumour purities into account.

In a symmetric distribution, the skewness value is zero, thus we find a mapping which minimises the skewness of region-wise tumour purities to adjust the tumour purity. Let $W = \{W_1, ..., W_K\}$ be parameters of the mapping for the estimated tumour purity in $K$ regions, $\delta = \{\delta_1, ..., \delta_K\}$. Assuming that the DMR $k$ includes $N$ reads, $\{r^k_1, ..., r^k_N\}$, the region-wise tumour purity $\delta_k$ is calculated as:

$$\delta_k = argmax_\delta \prod_{r^k_i\in\{r^k_1,...,r^k_N\}}\left[\delta P(r^k_i|cell\,type = Tumour)\right. \left. + (1-\delta)P(r^k_i|cell\,type = Normal)\right]. \quad (9)$$

The skewness of region-wise tumour purities can be calculated via the adjusted Fisher-Pearson standardised moment coefficient:

$$G_1(\delta) = \frac{m_3(\delta)\sqrt{K(K-1)}}{m_2(\delta)^{3/2}(K-2)}, \; m_t(\delta) = \frac{1}{K}\sum_{i=1}^{K}(\delta_i - \mu)^t \quad (10)$$

where $\mu$ is the sample mean of region-wise tumour purities, $\frac{1}{K}\sum_{i=1}^{K}\delta_i$. Therefore, the mapping parameters $W$ are optimised to minimise the skewness of region-wise tumour purities as follows:

$$\hat{W} = argmin_W G_1(W \circ \delta) = argmin_W G_1(\{W_1\delta_1, ..., W_K\delta_K\}) \quad (11)$$

where $W \circ \delta$ refers to the element-wise multiplication of two vectors $W$ and $\delta$. The expectation-maximisation (EM) algorithm is used to find the optimal mapping parameters $\hat{W}$. Once the best mapping is found, we assume that the distribution of region-wise estimates is symmetric and determine the final estimation of tumour purity as the mean value of mapped region-wise tumour purities:

$$\hat{\delta} = \frac{1}{K}\hat{W}^\top\delta. \quad (12)$$

The Fisher information indicates the amount of information about a model parameter carried by observed variables. In other words, the Fisher information is equivalent to an estimate of the model precision[22]. It is calculated as the variance of the derivative of the log-likelihood function with respect to the model parameter:

$$FI(\delta) = Var\left[\frac{\partial}{\partial\delta}\log L(\delta)\right]. \quad (13)$$

When the tumour purity adjustment is applied, the Fisher information cannot be calculated for the final estimation but in each region. Therefore, in this case, MethylBERT provides the Fisher information values as many as the number of selected DMRs.

## Data preparation

**Diffuse large B cell lymphoma WGBS data**. Diffuse large B cell lymphoma (DLBCL) and non-neoplastic B-cell WGBS data were downloaded from the Gene Expression Omnibus (GEO) database with the accession number GSE137880[39]. Eight samples from two DLBCL patients and eight non-neoplastic B cell samples from two donors were used for the experiments. From each subject, 4 samples were assigned for the training and validation dataset and the rest were used for

creating pseudo-bulks. All downloaded FastQ files were aligned with hg19 reference genome by Bismark 0.22.3[40] after trimming using TrimGalore 0.6.6[41], then duplicated reads were removed using picard Mark Duplicates 1.141. We followed the whole pipeline specified in our previous benchmarking study[13]. For Houseman's method, the read-level methylomes were converted into an array shape containing beta-values by Methrix[42]. Since Methrix requires a bedGraph file not a BAM file as an input, we converted the BAM files into a bedGraph file format using MethylDackel (https://github.com/dpryan79/MethylDackel).

**CRC and PDAC BS-seq data.** To train each MethylBERT model for ctDNA analysis derived from CRC and PDAC patients, we downloaded single-cell BS-seq of CRC cells with the GEO accession number GSE97693[43] and WGBS of PDAC tissue with GSE63123[44]. Only part of the CRC samples (341 cells) were downloaded as a training dataset. Similarly, we utilised only a subset of the PDAC data set involving seven samples collected from primary tumours in three different patients. During the data processing to convert the FastQ files to BAM files, we used TrimGalore 0.6.6 for adapter trimming and quality control. Afterwards, the samples were aligned with the hg19 reference genome using Bismark 0.22.3. Duplicate removal was done by picard Mark Duplicates 1.141.

**CRC, PDAC and control healthy ctDNA blood plasma data.** For ctDNA experiments, we downloaded targeted BS-seq of plasma samples from gastrointestinal cancer patients and controlled healthy donors with the GEO accession number GSE149438[28]. The downloaded dataset includes samples from 46 healthy donors, 74 PDAC patients, and 40 CRC patients in different cancer stages. We only used 44 PDAC patients whose cancer stage is clarified for the experiment. 32 healthy plasma samples were used to fine-tune the MethylBERT model and the rest was used as a comparison group in the tumour diagnosis analyses. We processed the data in FastQ files via trimming using TrimGalore, alignment to hg19 using Bismark and duplicate removal using picard Mark Duplicates. The samples were aligned with paired-end mode and the unmapped reads were re-aligned with single-end mode.

## DMR calling

Selecting informative regions with tumour-specific signals is vital in tumour purity estimation. For MethylBERT, we pre-selected DMRs to collect reads presenting informative methylation patterns. Tumour-specific DMRs were called by comparing tumour samples to non-tumour samples and the DSS package was used for the calling[45]. Parameters were set up as follows: delta value 0.2, P-value threshold 0.05, minimum number of CpGs 4, minimum length 50 bps and distance to merge 50 bps. In all analyses, we picked the top 100 DMRs based on the highest areaStat score. DSS performs a Wald test to identify differentially methylated loci and the areaStat score is calculated by summing all the test statistics up within each DMR. Therefore, a higher value of areaStat is likely to secure a larger number of CpGs in the region that are significantly differentially methylated between tumour and normal cell types.

## Read-level methylome simulation

Simulated read-level methylomes were used to mimic different scenarios of tumour-specific signals for the evaluation of methylation pattern classification. We selected 100 CpG islands with the highest number of CpGs as the regions where reads are sampled. For each region, two mean methylation levels need to be assigned for tumour and normal cell types. We sampled a tumour mean methylation value $d_i$ from a beta distribution whose $\beta$ parameter is fixed to 5 and assigned $1 - d_i$ to a normal mean methylation value. Then, read-level methylation patterns were sampled from a binomial distribution with the probability $1 - d_i$ and $d_i$ for normal and tumour cell types, respectively. The entire mechanism of sampling a read-level

methylome with $K$ CpGs, $\boldsymbol{m}^T = \{m^T{}_1, ..., m^T{}_K\}$ for tumour cell type and $\boldsymbol{m}^N = \{m^N{}_1, ..., m^N{}_K\}$ for normal cell type, is described as follows:

$$d_i \sim Beta(\alpha, \beta = 5), \tag{14}$$

$$\boldsymbol{m}^T \sim Binomial(n = K, p = d_i), \tag{15}$$

$$\boldsymbol{m}^N \sim Binomial(n = K, p = 1 - d_i) \tag{16}$$

where $\alpha$ and $\beta$ are two shape parameters of the beta distribution, while $n$ and $p$ are the number of trials and the success probability of one trial in the binomial distribution.

Four different $\alpha$ values, 0.1, 1.0, 2.0 and 3.0, were used in the beta distribution to model different complexities of methylation patterns. Since a larger alpha value increases the variance of the beta distribution with a fixed beta value, it is more likely that a lower tumour methylation value $d_i$ will be sampled. This makes a higher methylation value $1 - d_i$ assigned to normal cell type. The smaller gap between $d_i$ and $1 - d_i$ forms more complex methylation patterns between tumour and normal cell types (Supplementary Fig. 2).

When CpG-specific methylation patterns are sampled (Supplementary Fig. 2D), $d_i$ and $1 - d_i$ are assigned to the odd and even indices of CpGs. Therefore, in this simulation, the average methylation level between tumour and normal cell types do not significantly differ, but still have cell type-specific methylation patterns.

## Evaluation of read classification performance

The read classification performance evaluation includes an HMM-based algorithm. We designed the HMM to take methylation patterns as an observation, thus the observation has two categories: methylated and unmethylated CpGs. The hidden state is also a two-categorical variable with the assumption of whether the CpG is differentially methylated or not between tumour and normal cell types.

Although CancerDetector[15] and DISMIR[16] do not explicitly mention 'read classification', CancerDetector calculates $P(read|cell\ type = Tumour)$ whereas DISMIR has a 'd-score' that quantifies the chance that a given read is derived from tumour cells as interpreted by the authors. Therefore, we conducted read classification for those methods by assigning every read to the tumour class when the probability or $d$-score > 0.5.

In Fig. 2D, E and Supplementary Fig. 2, we evaluated MethylBERT read classification with different read coverage of the training set and the results show that MethylBERT can classify reads accurately in the case that the sample mean cannot represent the population mean well due to the low coverage. The standard error of the mean (SEM) was used for measuring how likely sample means of simulated read-level methylation patterns represent the real population mean. The SEM was calculated over the average methylation level of reads in each region.

## Prostate epithelium proportion analysis for the lymph node sample acquired from prostate cancer patients

We collected 15 lymph node samples from hormone-sensitive metastatic prostate cancer patients and generated both WGBS and 450K array data. The 450K data were processed following[46], while the WGBS data were processed following the DLBCL data preprocessing. For the array-based deconvolution methods[21,25–27], in-house prostate tumour and epithelial cell reference (as described in ref. 47) and additional cell types acquired from GEO (with the accession numbers GSE35069, GSE86258, GSE74877, GSE71837, GSE49667 and GSE87797) were used: monocytes, B cells, T cell, natural killer cells, granulocytes, fibroblasts, endothelium and mesenchymal stromal cells.

MethylBERT was trained with samples collected from the normal cell type atlas[23] (details are given in Supplementary

Table 4) not using any tumour reference data. The code is available at https://github.com/CompEpigen/wgbs_atlas_simulation.git. Instead of tumour reference data, we used four prostate epithelium samples, whereas we used two blood T cell, two blood B-cell, and one colon fibroblast samples for the normal cell-type label considering the composition of normal lymph node bulks. For DMRs, we used the top 250 hypomethylated regions in prostate epithelium cells provided by the atlas. After the fine-tuning, we performed prostate epithelium proportion estimation with the WGBS data from the lymphoma samples.

### Leukocyte subtypes deconvolution

We downloaded leukocyte WGBS samples collected from 23 healthy donors provided by the normal cell-type methylation atlas[23]. For the MethylBERT fine-tuning, we used two samples for each blood cell type as training data (details are given in Supplementary Table 5). Since the atlas only provides .pat file format, not raw sequence data, we converted the .pat files to read-level methylomes. The code is available on our GitHub repository (https://github.com/CompEpigen/wgbs_atlas_simulation.git).

We used 4 encoder layers and set all hyperparameters as described above, but increased the sequence length to 160. For the loss function, the focal loss[47] was used instead of the cross-entropy loss because multiple cell types make an imbalanced read distribution in terms of corresponding and non-corresponding cell types in each DMR. For example, in tumour purity estimation, tumour and normal cell types comprise roughly 50% of reads in each DMR in the case that both cell-type reference data have similar read coverage. Yet, if we want to perform five cell-type deconvolution, the corresponding cell type and non-corresponding cell types make up roughly 20% and 80% of the distribution in each DMR. This is called a class imbalance problem and often impedes deep neural network training. Focal loss was proposed to alleviate this problem by introducing a factor adjusting the loss value according to the misclassification. We followed the original implementation of the focal loss:

$$p = \sigma(x^c{}_r), \tag{17}$$

$$L_{focal-loss}(x^c{}_r) = \begin{cases} -\alpha \cdot (1-p)^\gamma \cdot log(p) & \text{if } c = \text{corresponding cell type} \\ -(1-\alpha) \cdot p^\gamma \cdot log(p) & \text{otherwise}, . \end{cases} \tag{18}$$

where $x^c{}_r$ is the final activation for the cell type $c$ for each read $r$. Although we used the softmax function in tumour purity estimation, here, we used the sigmoid function $\sigma(\cdot)$ following the original implementation. The authors explained that sigmoid operation shows better numerical stability. $\gamma$ and $\alpha$ are hyperparameters of the focal loss function and were set up as 2 and 0.1 for the leukocyte deconvolution.

After classifying sequence reads, we applied a new likelihood function to estimate the proportion of $C$ cell types, $\boldsymbol{\theta} = \{\theta_1, \ldots, \theta_C\}$:

$$L(\boldsymbol{\theta}) = \prod_{c=1}^{c} \prod_{r_i \in R_c} \left[ \theta_c P(r_i | cell\ type_c) + (1-\theta_c)(1 - P(r_i | cell\ type_c)) \right] \tag{19}$$

$$\hat{\boldsymbol{\theta}} = argmax_{\boldsymbol{\theta}} L(\boldsymbol{\theta}), \tag{20}$$

where $R_c$ is a group of reads classified into cell type $c$ which is written as $cell\ type_c$ in Eq. (19).

### Reporting summary

Further information on research design is available in the Nature Portfolio Reporting Summary linked to this article.

## Data availability

All data sets used in the study were downloaded from Gene Expression Omnibus (GEO). DLBCL WGBS data was downloaded with the accession number GSE137880, and ctDNA blood plasma samples (targeted BS-seq) were downloaded with the accession number GSE149438. The colorectal cancer (scBS-seq) and pancreatic cancer (WGBS) samples were downloaded with the accession numbers GSE97693 and GSE63123, respectively. All samples we downloaded from the normal cell atlas are available with the accession number GSE186458. Lymph node metastasis samples from prostate cancer patients are uploaded to the European Genome-Phenome Archive under the ID EGAS50000000806. Source data are provided with this paper.

## Code availability

The read-level methylation simulation code is available at https://github.com/CompEpigen/methylseq_simulation (https://doi.org/10.5281/zenodo.14025025). The normal cell-type atlas data processing code is available at https://github.com/CompEpigen/wgbs_atlas_simulation.git (https://doi.org/10.5281/zenodo.14025054). The MethylBERT code and package are available at https://github.com/CompEpigen/methylbert (https://doi.org/10.5281/zenodo.14025052) and https://pypi.org/project/methylbert/.

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

## Acknowledgements

We acknowledge extensive discussions from Verena Wolf (Saarland University), Christoph Plass (German Cancer Research Center), Lisa Barros de Andrade e Sousa and Marie Piraud (Helmholtz AI). We also acknowledge the German Cancer Research Center Genomics and Proteomics Core Facility for their technical support. Furthermore, we appreciate the support from funding conferred to the authors. Y.J. received support from the DKFZ-MOST scholarship (grant number CA191) and DKFZ CancerTRAX programs (grant number HIRS-0003). P.L. received support from KU Leuven BOF fund (starting grant, BOFZAP position). The lymph node WGBS data generation was funded by the ICGC EOPC (01KU1001 A) project on early-onset prostate cancer by the German Federal Ministry of Education and Research (BMBF). K.R. acknowledges the support of the BMBF within de.NBI/ELIXIR-DE (HD-HuB).

## Author contributions

Y.J. and P.L. conceptualised the study and developed the method. Y.J. implemented the code, performed the experiments, and drafted the original paper. P.L. and K.R. provided supervision and revised the paper. C.G., G.S. and T.S. provided the lymph node metastasis samples from prostate cancer patients and supported the corresponding analyses.

## Funding

## Competing interests

The authors declare no competing interests.
