## [Transparent Peer Review file · Nature Communications]

MethylBERT enables read-level DNA methylation pattern identification and tumour deconvolution using a Transformer-based model

Corresponding Author: Professor Pavlo Lutsik

Version 0:

Reviewer comments:

Reviewer #1

(Remarks to the Author)

Jeong et al. present a computational tool to leverage sequencing-based methylation measurements for pattern discovery and tumor deconvolution. Here the authors implement a 'bidirectional encoder representations from transformers' (BERT) model to encode read-level methylomes and classify methylation tagged sequencing reads into tumour or normal cell types. To assess predicted model precision Fisher information and estimation adjustment taking the region-wise tumour purity estimates into account is included in the implementation. The authors demonstrate the performance of the implementation, named methylBERT, with synthetic data, in silico pools of sequence-based datasets, bulk tumour and cfDNA datasets. Results of comparisons with existing tools are provided which demonstrate equivalent if not better performance of methylBERT when ground truth is known. A pip install of MethylBERT v0.0.2 code is provided.

Synopsis: Although this reviewer cannot speak to the novelty of applying BERT to the problem of the WGBS deconvolution, the availability of a benchmarked tool to enable accurate deconvolution of bulk WGBS datasets would be a significant utility to the community.

Critiques:

1. It would be helpful to understand in general terms, how the sequence datasets in particular are leveraged by methylBERT. The argument is made that sequence-based methylation measurements have particular advantages over analog array based Beta-values – but no comparisons are provided to reinforce this conjecture.
2. Can the authors comment on how methylBERT compares to the fragment-based deconvolution strategy employed by Loyfer et al. in the context of the methylation Atlas?
3. The details of how the transformation of primary fastq to fractional methylation calls could be improved and clarified. The authors point to a previous publication for the B-cell WGBS datasets, but appear to use a different workflow for BS-seq (should this also be WGBS?). It would be helpful to indicated how the sequence alignments were transformed after alignment and duplicate marking. Was a coverage threshold used to call methylation values? How did this differ for single cell methylation datasets - presumably this involved a different threshold for calling methylation values? Summary statistics on coverage, percent methylation, duplicates etc.. would also be helpful. For the public methyl-ATLAS datasets, did the authors use pre-processed methylation values – if so which ones?
4. What was the rationale for using single cell methylation datasets in this comparison – since by definition deconvolution should not apply. Were the single cells leveraged in some way, or pooled – this is not clear.
5. All datasets generated/ analyzed in this study should be made available (noting that “The lymph node WGBS data from prostate cancer patients are available upon request.”).
6. Can the author comment on the instability shown in Figure 2D (DISMIR) and Figure 4A (methylBERT). For DISMIR it seems to reach a coverage threshold?

7. Selected reference callouts appear to be incorrect (e.g. 44 on page 18, there may be others).

6. Can the authors provide run time and compute resources required to run methylBERT compared to other benchmarked algorithms.

(Remarks on code availability)

Reviewer #2

(Remarks to the Author)

Strengths:

MethylBERT pretraining analysis shows biologically relevant patterns are being learned during pretraining. This includes grouping all the starting nucleotides as well as kmers containing 'CG'. Just as DNABERT attention visualizations show the model is focusing on important, known motif in the DNA sequence, MethylBERT seems to also be picking up important features relevant to DNA methylation. Interestingly enough, pretraining on either human or mouse genomes does not affect read classification. Tests were done using both simulated data as well as on actual. Based on the simulation, MethylBERT performs better than all methods regardless of complexity using 150 and 500bp read lengths. MethylBERT also performs well with CpG specific methylation patterns, but does not perform better than DISMIR in the more complex read simulations, though the authors do not explain why. MethylBERT's ability to make accurate predictions even in low read coverage lends itself to application in ctDNA, since the read coverage for ctDNA is usually low, especially in the earlier stages of disease. MethylBERT also performed well using normal cell atlas data, which is also important clinically in case the patient reference sequence is not available.

Concerns:

1. Similar to DNABERT, creation of the kmers is via overlapping nucleotides, and the masked sequence prediction is on a contiguous sequence of these kmers. This may allow for the model to take a shortcut.
2. Provide a justification for choosing 3-mer sequences over longer sequences like 6-mer, as used in other models such as DNABERT. Are there specific advantages or limitations associated with using 3-mer in the context of DNA methylation pattern analysis that influenced this choice?
3. Absolute error comparisons in Figure 4 B and C do not include any statistical significance, though the paper states that the error is lower in the pre-trained MethylBERT model and performs better than other models. Figure 4 F shows the predicted methylation profile of a tumor and normal sequence. The authors state they are similar, but for me it does not look that similar. The authors may need to include a way to quantify the similarity between the methylation profiles or maybe not include the figure as it does not add to the story (MethylBERT's strength is predicting tumor vs normal, not generating the methylation profile).
4. What are the limitations regarding the maximum sequence length that can be input during the fine-tuning phase?

(Remarks on code availability)

Reviewer #3

(Remarks to the Author)

The paper presents MethylBERT, a novel Transformer-based model designed to identify DNA methylation patterns and perform tumour deconvolution from sequencing reads. By leveraging a Bidirectional Encoder Representations from Transformers (BERT) model, MethylBERT classifies methylation patterns and estimates tumour cell fractions in bulk samples with high accuracy. The model outperforms existing deconvolution methods and demonstrates its potential for early cancer diagnostics through the analysis of liquid biopsy samples. The code to perform the simulation presented in this paper, as well as the code and package, are shared via GitHub repositories. Both repositories look clean, and the README files are easy to follow.

In general, to my understanding, the term "deconvolution" refers to estimating several cell type fractions, while the method presented here performs a tumor purity estimation as it distinguishes only between two cell types: tumor and normal cells. Of course, as stated by the authors, the method can be extended to estimate other, possibly unknown, cell types, but this has not been demonstrated here. I believe that accomplishing this would require substantial modifications and validation of the presented method.

Line 59: The sentence: "MethylBERT uses a pre-trained BERT model to encode read-level methylomes and classify sequencing reads into tumour or normal cell types." might be a bit confusing, as it initially suggests that some publicly available pre-trained language model was used here.

Figure 4:

- A: How many data sets have been simulated? One for each tumour purity level?
- B: Are these boxplots calculated using all tumour purity levels?

- C and D: It is unclear to me which method has been used for 'no adjustment'.
- F: How exactly does MethyBERT estimate methylation beta values?

As also described by the authors, the implementation of Transformer models like MethyBERT can be computationally intensive, which may limit accessibility for some laboratories. What are the minimum requirements to perform inference with MethyBERT in a diagnostic setting?

(Remarks on code availability)

The code used for simulating the data as well as the code used to train the model and perform model inference are shared through two different GitHub repositories. The README files are easy to follow. The code for the MethyBERT model is also available as python module and can be easily installed using the pip package manager.

Version 1:

Reviewer comments:

Reviewer #1

(Remarks to the Author)

I would like to thank the authors for addressing my comments and for their patience in explaining how methyBERT leverages read-level data – this was not clear in my original read and significantly clarifies a number of questions raised in my original critique. The results, additional data analysis and edits included in the revision support the study's main conclusions and the authors have adequately addressed my remaining concerns. I would however request that all datasets included in this study be made available at the time of publication. EGA is of course acceptable and the EGA study and specific sample accessions can be listed if these data are also included in another study.

(Remarks on code availability)

Reviewer #3

(Remarks to the Author)

After carefully reviewing the manuscript and assessing the revisions made, I am pleased to report that the authors have adequately addressed all major concerns previously raised.

In light of these improvements, I find the article to be suitable for publication in its current form and recommend it for acceptance.

(Remarks on code availability)

RESPONSE TO REVIEWERS' COMMENTS

Reviewer #1 (Remarks to the Author):

Jeong et al. present a computational tool to leverage sequencing-based methylation measurements for pattern discovery and tumor deconvolution. Here the authors implement a 'bidirectional encoder representations from transformers' (BERT) model to encode read-level methylomes and classify methylation tagged sequencing reads into tumour or normal cell types. To assess predicted model precision Fisher information and estimation adjustment taking the region-wise tumour purity estimates into account is included in the implementation. The authors demonstrate the performance of the implementation, named methylBERT, with synthetic data, in silico pools of sequence-based datasets, bulk tumour and cfDNA datasets. Results of comparisons with existing tools are provided which demonstrate equivalent if not better performance of methylBERT when ground truth is known. A pip install of MethylBERT v0.0.2 code is provided.

Synopsis: Although this reviewer cannot speak to the novelty of applying BERT to the problem of the WGBS deconvolution, the availability of a benchmarked tool to enable accurate deconvolution of bulk WGBS datasets would be a significant utility to the community.

We would like to thank the reviewer for taking the time to read our manuscript and the valuable suggestions. We have addressed each critique/comment thoroughly and provided our responses point-by-point below.

Critiques:

1. It would be helpful to understand in general terms, how the sequence datasets in particular are leveraged by methylBERT. The argument is made that sequence-based methylation measurements have particular advantages over analog array based Beta-values – but no comparisons are provided to reinforce this conjecture.

There exist four major benefits of sequencing-based methylation profiling over array-based profiling with evidence from other previous work.

1. Sequencing-based profiling covers significantly more CpGs than array-based profiling. Array-based profiling technologies (EPIC array or 450K array) cover only a maximum of ~ 870,000 out of ~ 28 million CpG sites in the human genome. On the other hand, sequencing-based profiling technologies cover more than a million CpGs in the case of RRBS or (ideally) all CpGs in the case of WGBS. These comparisons are shown in Review Figure 1 from [Stirzaker et al., 2014].

Review Figure 1. Comparison of methylation profiling technologies from [Stirzaker et al., 2014]

2. Sequencing-based profiling preserves single-molecule-level information and consecutive methylation patterns on sequence reads. While array-based profiling provides only one beta-value (calculated by dividing methylated probe intensity by overall intensity) for each CpG in a sample, we can obtain multiple methylation patterns from sequence reads at one CpG site from sequencing-base profiling. Furthermore, methylomes on the same read preserve methylation patterns over neighbouring CpGs from the same cell. This information is not available in array-based profiling due to the beta-value computation done at each CpG independently. The consecutive read-level methylation patterns are particularly important for tumours where partially methylated domain and allele-specific methylation frequently occur [Do et al., 2020; Nishiyama et al., 2021]. These cannot be well-analysed via array-based profiling.
3. Array-based profiling could be limited in terms of sensitivity which is crucial for a small burden of tumour in the sample. The limited sensitivity is also caused by the calculation of beta-values where rare cell-type signals could be averaged out. This has been pointed out by [Moss et al., 2018; Loyfer et al., 2023] emphasising the necessity of sequencing-based profiling, particularly for cell-free DNA samples.
4. MethyBERT's use of methylation patterns instead of beta-values enables tumour purity estimation on genomic regions void of large methylation differences, which would be challenging to achieve for standard array-based deconvolution (see examples in Figure 2A-C).

Despite the major benefits of sequencing-based profiling, it is still necessary to choose the right computational method to ensure accurate estimation of tumour purity / cell-type

proportions from the data. In our previous study [Jeong et al., 2022], the benchmarking results indicate that not all deconvolution methods targeting sequencing-based data can leverage the benefits of read-level methylation patterns. Our manuscript shows that MethyBERT can achieve better tumour purity estimation performance compared to other methods by leveraging the benefits of sequencing-based data via multiple benchmarking experiments.

Specifically, MethyBERT leverages these benefits by using the BERT model and classifying individual sequence reads. The BERT model enables bidirectional learning which aligns with the nature of non-directional DNA sequence information. MethyBERT models the associations between 3-mer tokens (including their methylation patterns in the case of CpG) so that the model can recognise sequential methylation patterns on the read. Classification of individual sequence reads in MethyBERT makes the model robust to the sequencing depth and read coverage. This also enables precise detection of sequence reads derived from rare cell types, which is important for cell-free DNA analyses.

References

Stirzaker, Clare, et al. "Mining cancer methylomes: prospects and challenges." Trends in Genetics 30.2 (2014): 75-84.

Nishiyama, Atsuya, and Makoto Nakanishi. "Navigating the DNA methylation landscape of cancer." Trends in Genetics 37.11 (2021): 1012-1027.

Do, Catherine, et al. "Allele-specific DNA methylation is increased in cancers and its dense mapping in normal plus neoplastic cells increases the yield of disease-associated regulatory SNPs." Genome biology 21 (2020): 1-39.

Moss, Joshua, et al. "Comprehensive human cell-type methylation atlas reveals origins of circulating cell-free DNA in health and disease." Nature communications 9.1 (2018): 5068.

Loyfer, Netanel, et al. "A DNA methylation atlas of normal human cell types." Nature 613.7943 (2023): 355-364.

Jeong, Yunhee, et al. "Systematic evaluation of cell-type deconvolution pipelines for sequencing-based bulk DNA methylomes." Brief. Bioinformatics 23 (2022).

2. Can the authors comment on how methyBERT compares to the fragment-based deconvolution strategy employed by Loyfer et al. in the context of the methylation Atlas?

The fragment-based deconvolution method suggested by Loyfer et al. is fundamentally different to MethyBERT even though both methods primarily target BS-seq data. Loyfer et al.'s method assigns a methylation status (unmethylated/methylated/mixed) to sequence reads and then creates a vector of unmethylated fragment proportions for 1,232 markers (regions) to apply non-negative linear square optimisation. This approach is rather similar to array-based deconvolution: using averaged values of (un)methylation level and estimating cell-type proportions from a vector/array of the averaged values. As pointed out in our manuscript, such approaches cannot exploit the full advantages of sequencing-based data

since they lose single-molecule-level methylation patterns via average values. MethyBERT was developed to tackle this issue.

From a practical point of view, the MethyBERT implementation is more flexible than Loyfer et al.'s method implementation. Loyfer et al.'s method uses 1,232 designated cell type-specific unmethylation markers (25 markers for each cell type). Additional modifications of implementation would be necessary to introduce new cell types or regions. Especially, for cancer samples, Loyfer et al.'s method is not directly applicable due to cancer cell types not being included in their study. On the other hand, MethyBERT is designed for users to provide their cell type of interest as training (reference) data which gives more flexibility in experimental design.

Due to these practical reasons, we could not perform an equivalent tumour purity estimation with Loyfer et al.'s method for a direct comparison. However, we could estimate prostate epithelial cell proportions for lymph node metastasis samples from prostate cancer patients (Figure 4G). Although the result shows a positive correlation with the estimated tumour purity by Houseman's method (using tumour reference data), the MethyBERT result yields a better correlation with a lower mean absolute error.

However, one has to note that Loyfer et al.'s method is expected to have better running time given that their model has fewer parameters and the model does not need to be trained by users. (They provide an already trained model with target data consisting of cell types represented in their methylation atlas.) Consequently, Loyfer et al.'s method does not require reference data and can be used without GPU.

3. The details of how the transformation of primary fastq to fractional methylation calls could be improved and clarified. The authors point to a previous publication for the B-cell WGBS datasets, but appear to use a different workflow for BS-seq (should this also be WGBS?). It would be helpful to indicated how the sequence alignments were transformed after alignment and duplicate marking. Was a coverage threshold used to call methylation values? How did this differ for single cell methylation datasets - presumably this involved a different threshold for calling methylation values? Summary statistics on coverage, percent methylation, duplicates etc.. would also be helpful. For the public methyl-ATLAS datasets, did the authors use pre-processed methylation values – if so which ones?

We would like to point out that we have not transformed FastQ files to fractional methylation calls (also known as beta-values). MethyBERT is a method designed to exploit the benefits of raw read-level methylation patterns by not averaging (transforming) the methylation levels over reads which results in fractional methylation calls.

In order to obtain the read-level methylation patterns from a raw FastQ file, we performed read trimming (using TrimGarlore), alignment (using Bismark), and duplicate removal (using picard Mark Duplicates) as described in the subsections of "Data preparation" designated for each data set. However, our description might have confused the reviewer by not mentioning exact file formats, thus we improved the clarity of the explanation.

Nevertheless, transformed fractional methylation calls mimicking an array-based data format are required by Houseman's method included in our benchmarking. Here, we used Methrix [Mayakonda et al., 2020] and MethyIDackel (<https://github.com/dpryan79/MethyIDackel>) to convert the aligned read-level methylations into fractional values. The additional explanations about this processing are added in the "Diffuse large B cell lymphoma WGBS data" section for better understanding. For a fair comparison, we only used the methylation values from selected DMRs which were chosen based on their high coverage and CpG density, thus we did not specifically set any thresholds asked by the reviewer.

For the methylation atlas data, we used processed .pat files provided by the authors since the raw sequence reads (FastQ or BAM files) are not publicly available. However, the .pat files do not exactly have read information, thus we implemented new preprocessing codes to extract read-level methylomes from the .pat files. The code is available on Github (https://github.com/CompEpigen/wgbs_atlas_simulation.git). The samples of methylation atlas that we used are listed in Supplementary Tables 4 and 5.

Reference

Mayakonda, Anand, et al. "Methrix: an R/Bioconductor package for systematic aggregation and analysis of bisulfite sequencing data." *Bioinformatics* 36.22-23 (2020): 5524-5525.

4. What was the rationale for using single cell methylation datasets in this comparison – since by definition deconvolution should not apply. Were the single cells leveraged in some way, or pooled – this is not clear.

We would like to note that single-cell data has not been used for any deconvolution/tumour purity estimation but for training MethyBERT applied to the CRC patients' ctDNA analysis. These single-cell data were acquired from primary CRC tumour tissues. With single-cell methylation profiles, MethyBERT can learn the reference methylation patterns for the CRC compartment. However, we would like to clarify that we used targeted BS-seq data of blood plasma for the ctDNA samples where we estimated the tumour burden.

To be more clear, MethyBERT requires "reference data" collected from tumours and normal samples to train the model. For CRC results, fortunately, as explained, we found single-cell BS-seq data for the tumour samples (GSE97693). Technically, single-cell data is preferred for training MethyBERT because it provides better resolutions of tumour methylation patterns. WGBS/targeted BS-seq data from tumour bulks can always contain some non-tumour cells (e.g., immune cells), which can introduce undesired sources of variation in the training data sets. However, tumour scBS-seq data is not very common thus we had to use WGBS/targeted BS-seq data sets for other experiments (B-cell lymphoma and PDAC ctDNA experiments).

5. All datasets generated/ analyzed in this study should be made available (noting that "The lymph node WGBS data from prostate cancer patients are available upon request.").

Since the lymph node WGBS data originates from patient samples, we are unfortunately not able to share the full raw sequence alignments with unrestricted access according to the data protection law. However, we generated preprocessed data from the lymph node WGBS

data by selecting reads overlapping with DMRs used in the manuscript. We provide it as Supplementary Data so the readers can reconstruct the results from the manuscript. The raw data is being uploaded to EGA in a framework of a larger study that is being prepared.

6. Can the author comment on the instability shown in Figure 2D (DISMIR) and Figure 4A (MethylBERT). For DISMIR it seems to reach a coverage threshold?

We performed further experiments for the DISMIR's sudden performance drop with a high read coverage in Figure 2D. After training DISMIR five times independently with the same data set, we found that DISMIR training is not very robust to high read coverage, meaning that the model randomly showed a low performance (Supplementary Figure 10B). On the other hand, MethylBERT training was performed robustly with a very small confidence interval over five independent trainings (Supplementary Figure 10A). For a reasonable performance comparison, we replaced the DISMIR result with the best-trained one in Figure 2D and addressed the robustness issue in the manuscript.

Concerning the instability in Supplementary Figure 11A (please note that we moved Figure 4A to Supplementary Figure 11A), we would like to point out that the lowest absolute error in the MethylBERT results (bulk whose tumour purity is 0.579) is overemphasised by the log-scaled y-axis. As shown in Supplementary Figure 11B (estimated tumour purity plotted without log-scale), we show that all estimated purities by MethylBERT are very close to the ground-truth line, not having an outlier in the line plot.

7. Selected reference callouts appear to be incorrect (e.g. 44 on page 18, there may be others).

We thank the reviewers for pointing out the mistakes in the references. We carefully went through the whole manuscript and fixed the incorrect reference callouts.

6. Can the authors provide run time and compute resources required to run methylBERT compared to other benchmarked algorithms.

Since MethylBERT is a Transformer-based model which generally has a significantly higher number of parameters than other deep learning models like recurrent neural networks (RNNs), more running time and computing resources are usually required than for other methods to analyse the same data. For example, during each experiment for Figure 2D, MethylBERT ran for ~ 1.5 hours with two GPUs (NVIDIA V100 32GB), whereas DISMIR running time was ~ 45 minutes with one GPU. (However, this was measured only for the running time comparison between MethylBERT and DISMIR using simulated reads. Please find the additional explanation about the MethylBERT running time for real tumour bulk data in Methods and Supplementary Figure 9.) Houseman's method and CancerDetector do not require GPUs for computation and the running time was far shorter than one hour. However, we would like to emphasise that the running time and computing resource requirements highly depend on the training strategy. For example, users can reduce the running time by using more GPUs with a larger batch size in the MethylBERT training hyperparameters.

In order to compensate for the computational burden, we implemented mixed precision in MethylBERT. Mixed precision replaces some 32-bit floating point type of variables with 16-bit

floating point type to decrease memory consumption. Broad sorts of processors including NVIDIA GPUs tend to fasten their operations with mixed precision. Therefore, it helps reduce computational resources (running time and CPU/GPU memory) required by MethyBERT.

Reviewer #2 (Remarks to the Author):

Strengths:

MethyBERT pretraining analysis shows biologically relevant patterns are being learned during pretraining. This includes grouping all the starting nucleotides as well as kmers containing 'CG'. Just as DNABERT attention visualizations show the model is focusing on important, known motif in the DNA sequence, MethyBERT seems to also be picking up important features relevant to DNA methylation. Interestingly enough, pretraining on either human or mouse genomes does not affect read classification. Tests were done using both simulated data as well as on actual. Based on the simulation, MethyBERT performs better than all methods regardless of complexity using 150 and 500bp read lengths. MethyBERT also performs well with CpG specific methylation patterns, but does not perform better than DISMIR in the more complex read simulations, though the authors do not explain why. MethyBERT's ability to make accurate predictions even in low read coverage lends itself to application in ctDNA, since the read coverage for ctDNA is usually low, especially in the earlier stages of disease. MethyBERT also performed well using normal cell atlas data, which is also important clinically in case the patient reference sequence is not available.

We appreciate the positive review and worthwhile additions to our study made by the reviewer. We have thoroughly looked through the concerns and suggestions, and made a response for each below.

Regarding the performance difference between MethyBERT and DISMIR for the more complex read simulations (Figure 2C), we believe that DISMIR achieved exceptionally better performance than other methods because the model takes a very short read length, 66 bps in the implementation. Thus, the input data needs to be trimmed for DISMIR and we split our data into the required sequence length (rather than trimming reads off) to exploit the read-level methylation patterns we have in the data set. Although, in practice, a shorter length of sequences is disadvantageous for read-level methylome analysis due to the loss of information, it can still increase the training performance of neural networks for simulated data by decreasing the complexity of methylation patterns. Therefore, we think that the performance of DISMIR was higher in the case of complex read simulations.

Concerns:

1. Similar to DNABERT, creation of the kmers is via overlapping nucleotides, and the masked sequence prediction is on a contiguous sequence of these kmers. This may allow for the model to take a shortcut.

As the reviewer described, MethyBERT creates k-mer tokens out of the DNA sequences and uses a masked language model for the pre-training. In the written concern, the reviewer mentioned that this approach “allows for the model to take a shortcut” but we could not find what kind of shortcuts the reviewer means. Therefore, here, we describe the benefits of using k-mer tokens for MethyBERT which can be considered a shortcut to efficient training.

K-mer tokenisation makes a larger vocabulary size (number of unique tokens) for the model than using 4 nucleotides as tokens. For example, 3-mer tokenisation as used in MethyBERT has a $4^3 = 64$ vocabulary size which is much larger than the vocabulary size of 4 when we use nucleotides as tokens. In Transformer-based model training, it is crucial to have a large vocabulary size because a larger vocabulary size leads to more parameters in the embedding layers which increases model complexity as well as enables the model to handle a larger variability within the data [Zhao et al., 2022; Takese et al., 2024]. Thus, we used k-mer tokenisation for the input data processing to ensure sufficient model complexity and variability within the data for MethyBERT pre-training to recognise complex DNA sequential patterns in (human) genomes.

References

Takase, Sho, et al. "Large Vocabulary Size Improves Large Language Models." arXiv preprint arXiv:2406.16508 (2024).

Zhao, Tiancheng, et al. "Omdet: Language-aware object detection with large-scale vision-language multi-dataset pre-training." arXiv preprint arXiv:2209.05946 (2022).

2. Provide a justification for choosing 3-mer sequences over longer sequences like 6-mer, as used in other models such as DNABERT. Are there specific advantages or limitations associated with using 3-mer in the context of DNA methylation pattern analysis that influenced this choice?

When it comes to choosing the number k in k-mer sequences, we focused on the information density of the final k-mer sequence and the implementation of CpG methylation patterns on top of the k-mer sequences.

The authors of DNABERT reported that DNABERT with $k=3, 4, 5,$ and 6 showed very similar performance. Using $k=6$ for BS-seq data sets could cause a major drawback for MethyBERT due to the read length being much shorter than DNA sequences used in the DNABERT paper. When the read length is 150 bps which is the most common for BS-seq data, the final overlapping k-mer sequence length is 148 for $k=3$ and 145 for $k=6$. Furthermore, when the masked language model is applied during the pre-training, k neighbouring tokens need to be masked to prevent the information leak from neighbouring tokens to the prediction of the targeted masked token. Consequently, using $k=6$ results in more masked tokens in a shorter sequence compared to $k=3$. This unnecessarily increases sparsity in the data which complicates the training. The sparsity within each data point is especially critical when the input sequence length is short like MethyBERT. Therefore, we decided to use $k=3$ primarily for the explained two reasons: different numbers of k do not make a significant performance difference according to the DNABERT paper and we can provide information-denser data to the model with $k=3$.

Another practical benefit of using $k=3$ (technically any odd numbers) is that it brings an intuitive design of incorporating CpG methylation patterns with the k -mer sequences. We assign a methylation pattern to each k -mer token when a CpG-context cytosine is the middle nucleotide of the token. However, if we use an even number for k , there is no unique middle nucleotide which makes the assignment of methylation pattern more ambiguous.

3. Absolute error comparisons in Figure 4 B and C do not include any statistical significance, though the paper states that the error is lower in the pre-trained MethylBERT model and performs better than other models. Figure 4 F shows the predicted methylation profile of a tumor and normal sequence. The authors state they are similar, but for me it does not look that similar. The authors may need to include a way to quantify the similarity between the methylation profiles or maybe not include the figure as it does not add to the story (MethylBERT's strength is predicting tumor vs normal, not generating the methylation profile).

Please note that the Figure number has changed from “Figures 4B and C” to “Figures 4A and B” in the new version of the manuscript.

We added the median value for each method's result and p-value via paired t-test in Figures 4A and B, and Supplementary Tables 2 and 3.

Regarding the reviewer's concern about Figure 4F (in the old version of the manuscript), we agree with the reviewer's opinion that the similarity needs to be quantified. After further analyses, we realised the difficulty of thorough similarity quantification for some bulk samples due to their low read coverage. For example, in the bulk whose tumour purity is 0.088, the number of reads originating from the tumour is too low and many CpGs are not covered for the tumour methylation pattern reconstruction which results in an imprecise similarity quantification. Thus, we excluded these results from the manuscript as suggested by the reviewer, and instead extended the analysis of DMR-wise methylation level dissection (Figure 4D and E, and Supplementary Figures 5 and 13).

4. What are the limitations regarding the maximum sequence length that can be input during the fine-tuning phase?

In theory, as a Transformer-based model, MethylBERT does not have a limit on input sequencing length. However, in practice MethylBERT restricts the length of input fragments because of limited computational resources (running time and CPU/GPU memory), meaning that the more computational resources are available, the longer the sequence can be handled by MethylBERT. With our computational resources, we used MethylBERT for a maximum of 500 bps sequence length.

Nevertheless, the implementation of the MethylBERT model makes the sequence length in fine-tuning upper-bound by the sequence length used in pre-training. MethylBERT uses absolute position encoding (following the design of the original BERT). In the absolute position encoding, the order index of the input sequence tokens is assigned to the position values and the embedding layer converts the value into a new embedding space. Since the model architecture established during pre-training cannot be changed during fine-tuning, the currently available pre-trained MethylBERT models (uploaded on

<https://huggingface.co/hanyangji>) can handle a maximum of 512 bps of reads which is the sequencing length used for the pre-training.

However, for an extension of MethyBERT for long-read sequencing data as mentioned in the manuscript, the improvement of position encoding can be done with more studies. [Kazemnejad et al., 2024] argued that Transformers without position encoding outperform the ones with position encoding at sequence length generalisation, thus we can also consider the removal of position encoding if the performance is preserved without the encoding.

Reference

Kazemnejad, Amirhossein, et al. "The impact of positional encoding on length generalization in transformers." *Advances in Neural Information Processing Systems* 36 (2024).

Reviewer #3 (Remarks to the Author):

The paper presents MethyBERT, a novel Transformer-based model designed to identify DNA methylation patterns and perform tumour deconvolution from sequencing reads. By leveraging a Bidirectional Encoder Representations from Transformers (BERT) model, MethyBERT classifies methylation patterns and estimates tumour cell fractions in bulk samples with high accuracy. The model outperforms existing deconvolution methods and demonstrates its potential for early cancer diagnostics through the analysis of liquid biopsy samples. The code to perform the simulation presented in this paper, as well as the code and package, are shared via GitHub repositories. Both repositories look clean, and the README files are easy to follow.

We are very grateful to the reviewer for spending the time to read our manuscript and providing meaningful comments we need to further address. We have answered the reviewer's comment point-by-point in the following.

In general, to my understanding, the term "deconvolution" refers to estimating several cell type fractions, while the method presented here performs a tumor purity estimation as it distinguishes only between two cell types: tumor and normal cells. Of course, as stated by the authors, the method can be extended to estimate other, possibly unknown, cell types, but this has not been demonstrated here. I believe that accomplishing this would require substantial modifications and validation of the presented method.

We appreciate the author's clarification of terminologies. We agree that the word "deconvolution" more refers to cell-type fraction estimation, thus we have replaced it with "tumour purity estimation" in the manuscript.

At the same time, we updated the MethyBERT implementation. MethyBERT is now available for cell-type deconvolution. We used the focal loss and a new likelihood function to

extend the MethyBERT model to multiple cell-type proportion estimation. Please find the details in Methods in the manuscript.

The extended MethyBERT was applied to 23 leukocyte bulk samples and the deconvolution results were compared to the results given by Loyfer et al.'s method (Figure 4H in the manuscript). For all five major cell types, the correlation between the two estimates is strongly positive showing the applicability of MethyBERT as a cell-type deconvolution method.

Reference

Loyfer, Netanel, et al. "A DNA methylation atlas of normal human cell types." *Nature* 613.7943 (2023): 355-364.

Line 59: The sentence: "MethyBERT uses a pre-trained BERT model to encode read-level methylomes and classify sequencing reads into tumour or normal cell types." might be a bit confusing, as it initially suggests that some publicly available pre-trained language model was used here.

We changed "pre-trained BERT model" to "modified BERT model" to avoid confusion. We used the BERT architecture with modifications for DNA sequences and methylation patterns, however, we have not used any publicly available pre-trained models since we pre-trained the model by ourselves. We would like to note that the pre-trained models are available on our HuggingFace repository (<https://huggingface.co/hanyangji>).

Figure 4:

• A: How many data sets have been simulated? One for each tumour purity level?

(Please note that this figure has been moved to Supplementary Figure 11A.)

We simulated a total of 20 bulk samples and, as the reviewer described, one bulk was assigned for each tumour purity level.

• B: Are these boxplots calculated using all tumour purity levels?

(Please note that this figure has been moved to Figure 4A.)

Yes, the boxplot for each model includes 20 bulk samples from all tumour purity levels.

• C and D: It is unclear to me which method has been used for 'no adjustment'.

(Please note that these figures have been moved to Figure 4B and Supplementary Figure 12.)

The label "no adjustment" indicates the estimated final tumour purity without any adjustment methods (neither MethyBERT nor CancerDetector adjustment methods are applied). In this case, we directly estimate the final tumour purity using the likelihood estimation (Line 598-599).

• F: How exactly does MethyBERT estimate methylation beta values?

We apologise for the potential confusion made in this section. MethyBERT does not estimate methylation beta-values but allows users to do a post-hoc reconstruction of methylation beta-values (and region-wise methylation levels) based on the read classification results. At each CpG site, we averaged the methylation pattern over all classified reads in respective cell types. However, the CpG-wise beta-value reconstruction results have been excluded from the manuscript due to another reviewer's concern about the clarity of the results that we agree with.

Instead, we have performed further analyses on region-wise methylation level dissection analyses (Figures 4D and E, and Supplementary Figures 5, 13 and 14). The results confirm that reconstructed methylation levels by averaging methylation levels of classified reads in each cell type accurately represent the reference cell type-specific methylation patterns.

As also described by the authors, the implementation of Transformer models like MethyBERT can be computationally intensive, which may limit accessibility for some laboratories. What are the minimum requirements to perform inference with MethyBERT in a diagnostic setting?

The training time of deep neural networks, especially large models like Transformers, highly depends on training strategy, thus it is hard to give an exact number for the minimum requirements of computational resources. In particular, we added a multiple-GPUs computation feature within the MethyBERT implementation for faster training, which makes the running time harder to estimate due to the data distributed on multiple devices. Nonetheless, we can provide guidance for training time related to computational resources.

For a diagnostic setting, the user needs one-time fine-tuning of MethyBERT with their training (reference) data. This requires sufficiently long training until the model converges at the ideal performance. Supplementary Figure 9 gives training time for the MethyBERT model fine-tuning with different numbers of encoder layers (determining the model size) by different numbers of GPUs (Nvidia V100 SXM2 32GB). We used approximately 541,000 reads as a training set as well as 135,000 reads as a validation set. The training time got longer with GPUs more than four because the given batch size (we assigned 600 for these experiments) is too small to reach the most efficient training time for six or eight GPUs. With a higher batch size, users can reduce the training time.

After the one-time fine-tuning, users can run MethyBERT tumour purity estimation for bulk samples independently. This requires much less computational resources. When we used one Nvidia V100 SXM2 32GB for one CRC circulating tumour DNA sample from Figure 5C, it took less than 5 minutes to estimate tumour proportion (batch size was set up as 256). When we ran the same experiment but without a GPU, it took 40-45 mins. This information has been added to Methods.

Reviewer #3 (Remarks on code availability):

The code used for simulating the data as well as the code used to train the model and perform model inference are shared through two different GitHub repositories. The README files are easy to follow. The code for the MethyBERT model is also available as python module and can be easily installed using the pip package manager.

We appreciate the reviewer's positive comments on the code availability

RESPONSE TO REVIEWERS' COMMENTS

Reviewer #1 (Remarks to the Author):

I would like to thank the authors for addressing my comments and for their patience in explaining how methylBERT leverages read-level data – this was not clear in my original read and significantly clarifies a number of questions raised in my original critique. The results, additional data analysis and edits included in the revision support the study's main conclusions and the authors have adequately addressed my remaining concerns. I would however request that all datasets included in this study be made available at the time of publication. EGA is of course acceptable and the EGA study and specific sample accessions can be listed if these data are also included in another study.

We would like to thank the reviewer for taking the time to read our responses and for the positive opinions. When it comes to the datasets that the reviewer requested to upload, we are currently arranging the EGA upload and it is expected to be available at the time of publication.

Reviewer #3 (Remarks to the Author):

After carefully reviewing the manuscript and assessing the revisions made, I am pleased to report that the authors have adequately addressed all major concerns previously raised.

In light of these improvements, I find the article to be suitable for publication in its current form and recommend it for acceptance.

We appreciate the reviewer's positive remarks and their time to review our manuscript.